# DIVERSE PROJECTION ENSEMBLES
# FOR DISTRIBUTIONAL REINFORCEMENT LEARNING

**Moritz A. Zanger**     **Wendelin Böhmer**     **Matthijs T. J. Spaan**
Delft University of Technology, The Netherlands
{m.a.zanger, j.w.bohmer, m.t.j.spaan}@tudelft.nl

## ABSTRACT

In contrast to classical reinforcement learning (RL), distributional RL algorithms aim to learn the distribution of returns rather than their expected value. Since the nature of the return distribution is generally unknown a priori or arbitrarily complex, a common approach finds approximations within a set of representable, parametric distributions. Typically, this involves a *projection* of the unconstrained distribution onto the set of simplified distributions. We argue that this projection step entails a strong inductive bias when coupled with neural networks and gradient descent, thereby profoundly impacting the generalization behavior of learned models. In order to facilitate reliable uncertainty estimation through diversity, we study the combination of several different projections and representations in a distributional ensemble. We establish theoretical properties of such *projection ensembles* and derive an algorithm that uses ensemble disagreement, measured by the average 1-Wasserstein distance, as a bonus for deep exploration. We evaluate our algorithm on the behavior suite benchmark and VizDoom and find that diverse projection ensembles lead to significant performance improvements over existing methods on a variety of tasks with the most pronounced gains in directed exploration problems.

## 1 INTRODUCTION

In reinforcement learning (RL), agents interact with an unknown environment, aiming to acquire policies that yield high cumulative rewards. In pursuit of this objective, agents must engage in a trade-off between information gain and reward maximization, a dilemma known as the exploration/exploitation trade-off. In the context of model-free RL, many algorithms designed to address this problem efficiently rely on a form of the *optimism in the face of uncertainty* principle (Auer, 2002) where agents act according to upper confidence bounds of value estimates. When using high-capacity function approximators (e.g., neural networks) the derivation of such confidence bounds is non-trivial. One popular approach fits an ensemble of approximations to a finite set of observations (Dietterich, 2000; Lakshminarayanan et al., 2017). Based on the intuition that a set of parametric solutions explains observed data equally well but provides diverse predictions for unobserved data, deep ensembles have shown particularly successful at quantifying uncertainty for novel inputs. An exploring agent may, for example, seek to reduce this kind of uncertainty by visiting unseen state-action regions sufficiently often, until ensemble members converge to almost equal predictions. This notion of reducible uncertainty is also known as *epistemic uncertainty* (Hora, 1996; Der Kiureghian and Ditlevsen, 2009).

A concept somewhat orthogonal to epistemic uncertainty is *aleatoric uncertainty*, that is the uncertainty associated with the inherent irreducible randomness of an event. The latter is the subject of the recently popular *distributional* branch of RL (Bellemare et al., 2017), which aims to approximate the distribution of returns, as opposed to its mean. While distributional RL naturally lends itself to risk-sensitive learning, several results show significant improvements over classical RL even when distributions are used only to recover the mean (Bellemare et al., 2017; Dabney et al., 2018b; Rowland et al., 2019; Yang et al., 2019; Nguyen-Tang et al., 2021). In general, the probability distribution of the random return may be arbitrarily complex and difficult to represent, prompting many recent advancements to rely on novel methods to *project* the unconstrained return distribution onto a set of representable distributions.

In this paper, we study the combination of different *projections* and *representations* in an ensemble of distributional value learners. In this setting, agents who seek to explore previously unseen states and

actions can recognize such novel, out-of-distribution inputs by the diversity of member predictions: through learning, these predictions align with labels in frequently visited states and actions, while novel regions lead to disagreement. For this, the individual predictions for unseen inputs, hereafter also referred to as generalization behavior, are required to be sufficiently diverse. We argue that the projection step in distributional RL imposes an inductive bias that leads to such diverse generalization behaviors when joined with neural function approximation. We thus deem distributional projections instrumental to the construction of diverse ensembles, capable of effective separation of epistemic and aleatoric uncertainty. To illustrate the effect of the projection step in the function approximation setting, Fig. 1 shows a toy regression problem where the predictive distributions differ visibly for inputs $x$ not densely covered by training data depending on the choice of projection.

Our main contributions are as follows:

(1) We introduce distributional *projection ensembles* and analyze their properties theoretically. In our setting, each model is iteratively updated toward the projected mixture over ensemble return distributions. We describe such use of distributional ensembles formally through a *projection mixture operator* and establish several of its properties, including contractivity and residual approximation errors.

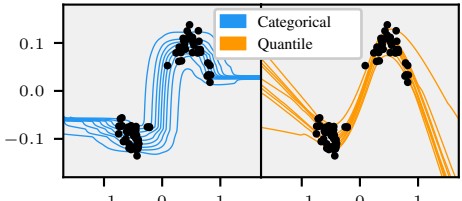

Figure 1: Toy 1D-regression: Black dots are training data with inputs $x$ and labels $y$. Two models have been trained to predict the distribution $p(y|x)$ using a categorical projection (l.h.s.) and a quantile projection (r.h.s.). We plot contour lines for the $\tau = [0.1, ..., 0.9]$ quantiles of the predictive distributions over the interval $x \in [-1.5, 1.5]$.

(2) When using shared distributional temporal difference (TD) targets, ensemble disagreement is biased to represent distributional TD errors rather than errors w.r.t. the true return distribution. To this end, we derive a *propagation* scheme for epistemic uncertainty that relates absolute deviations from the true value function to distributional TD errors. This insight allows us to devise an optimism-based exploration algorithm that leverages a learned bonus for directed exploration.

(3) We implement these algorithmic elements in a deep RL setting and evaluate the resulting agent on the behavior suite (Osband et al., 2020), a benchmark collection of 468 environments, and a set of hard exploration problems in the visual domain *VizDoom* (Kempka et al., 2016). Our experiments show that *projection ensembles* aid reliable uncertainty estimation and exploration, outperforming baselines on most tasks, even when compared to significantly larger ensemble sizes.

## 2 RELATED WORK

Our work builds on a swiftly growing body of literature in distributional RL (Morimura et al., 2010; Bellemare et al., 2017). In particular, several of our theoretical results rely on works by Rowland et al. (2018) and Dabney et al. (2018b), who first provided contraction properties with categorical and quantile projections in distributional RL respectively. Numerous recently proposed algorithms (Dabney et al., 2018a; Rowland et al., 2019; Yang et al., 2019; Nguyen-Tang et al., 2021) are based on novel representations and projections, typically with an increased capacity to represent complex distributions. In contrast to our approach, however, these methods have no built-in functionality to estimate epistemic uncertainty. To the best of our knowledge, our work is the first to study the combination of different projection operators and representations in the context of distributional RL.

Several works, however, have applied ensemble techniques to distributional approaches. For example, Clements et al. (2019), Eriksson et al. (2022), and Hoel et al. (2023) use ensembles of distributional models to derive aleatoric and epistemic risk measures. Lindenberg et al. (2020) use an ensemble of agents in independent environment instances based on categorical models to drive performance and stability. Jiang et al. (2024) leverage quantile-based ensembles to drive exploration in contextual MDPs, while Nikolov et al. (2019) combine a deterministic Q-ensemble with a distributional categorical model for information-directed sampling. In a broader sense, the use of deep ensembles for value estimation and exploration is widespread (Osband et al., 2016; 2019; Flennerhag et al., 2020; Fellows et al., 2021; Chen et al., 2017). A notable distinction between such algorithms is whether ensemble members are trained independently or whether joint TD backups are used. Our work falls into the latter category which typically requires a propagation mechanism to estimate value uncertainty rather

than uncertainty in TD targets (Janz et al., 2019; Fellows et al., 2021; Moerland et al., 2017). Our proposed propagation scheme establishes a temporal consistency between *distributional* TD errors and errors w.r.t. the true return distribution. In contrast to the related uncertainty Bellman equation (O'Donoghue et al., 2018), our approach applies to the distributional setting and devises uncertainty propagation from the perspective of error decomposition, rather than posterior variance.

## 3 BACKGROUND

Throughout this work, we consider a finite Markov Decision Process (MDP) (Bellman, 1957) of the tuple $(\mathcal{S}, \mathcal{A}, \mathcal{R}, \gamma, P, \mu)$ as the default problem framework, where $\mathcal{S}$ is the finite state space, $\mathcal{A}$ is the finite action space, $\mathcal{R} : \mathcal{S} \times \mathcal{A} \to \mathscr{P}(\mathbb{R})$ is the immediate reward distribution, $\gamma \in [0, 1]$ is the discount factor, $P : \mathcal{S} \times \mathcal{A} \to \mathscr{P}(\mathcal{S})$ is the transition kernel, and $\mu : \mathscr{P}(\mathcal{S})$ is the start state distribution. Here, we write $\mathscr{P}(\mathcal{X})$ to indicate the space of probability distributions defined over some space $\mathcal{X}$. Given a state $S_t$ at time $t$, agents draw an action $A_t$ from a stochastic policy $\pi : \mathcal{S} \to \mathscr{P}(\mathcal{A})$ to be presented the random immediate reward $R_t \sim \mathcal{R}(\cdot | S_t, A_t)$ and the successor state $S_{t+1} \sim P(\cdot | S_t, A_t)$. Under policy $\pi$ and transition kernel $P$, the discounted return is a random variable given by the discounted cumulative sum of random rewards according to $Z^\pi(s, a) = \sum_{t=0}^{\infty} \gamma^t R_t$, where $S_0 = s, A_0 = a$. Note that our notation will generally use uppercase letters to indicate random variables. Furthermore, we write $\mathcal{D}(Z^\pi(s, a)) \in \mathscr{P}(\mathbb{R})$ to denote the distribution of the random variable $Z^\pi$(s,a), that is a state-action-dependent distribution residing in the space of probability distributions $\mathscr{P}(\mathbb{R})$. For explicit referrals, we label this distribution $\eta^\pi(s, a) = \mathcal{D}(Z^\pi(s, a))$. The expected value of $Z^\pi(s, a)$ is known as the state-action value $Q^\pi(s, a) = \mathbb{E}[Z^\pi(s, a)]$ and adheres to a temporal consistency condition described by the Bellman equation (Bellman, 1957)

$$Q^\pi(s, a) = \mathbb{E}_{P,\pi}[R_0 + \gamma Q^\pi(S_1, A_1) | S_0 = s, A_0 = a], \tag{1}$$

where $\mathbb{E}_{P,\pi}$ indicates that successor states and actions are drawn from $P$ and $\pi$ respectively. Moreover, the Bellman operator $T^\pi Q(s, a) := \mathbb{E}_{P,\pi}[R_0 + \gamma Q(S_1, A_1) | S_0 = s, A_0 = a]$ has the unique fixed point $Q^\pi(s, a)$.

### 3.1 DISTRIBUTIONAL REINFORCEMENT LEARNING

The *distributional* Bellman operator $\mathcal{T}^\pi$ (Bellemare et al., 2017) is a probabilistic generalization of $T^\pi$ and considers return distributions rather than their expectation. For notational convenience, we first define $P^\pi$ to be the transition operator according to

$$P^\pi Z(s, a) :\overset{D}{=} Z(S_1, A_1), \qquad \text{where} \qquad S_1 \sim P(\cdot | S_0 = s, A_0 = a), \quad A_1 \sim \pi(\cdot | S_1), \tag{2}$$

and $\overset{D}{=}$ indicates an equality in distributional law (White, 1988). In this setting, the distributional Bellman operator is defined as

$$\mathcal{T}^\pi Z(s, a) :\overset{D}{=} R_0 + \gamma P^\pi Z(s, a). \tag{3}$$

Similar to the classical Bellman operator, the distributional counterpart $\mathcal{T}^\pi : \mathscr{P}(\mathbb{R})^{\mathcal{S} \times \mathcal{A}} \to \mathscr{P}(\mathbb{R})^{\mathcal{S} \times \mathcal{A}}$ has the unique fixed point $\mathcal{T}^\pi Z^\pi = Z^\pi$, that is the true return distribution $Z^\pi$. In the context of iterative algorithms, we will also refer to the identity $\mathcal{T}^\pi Z(s, a)$ as a bootstrap of the distribution $Z(s, a)$. For the analysis of many properties of $\mathcal{T}^\pi$, it is helpful to define a distance metric over the space of return distributions $\mathscr{P}(\mathbb{R})^{\mathcal{S} \times \mathcal{A}}$. Here, the supremum $p$-Wasserstein metric $\bar{w}_p : \mathscr{P}(\mathbb{R})^{\mathcal{S} \times \mathcal{A}} \times \mathscr{P}(\mathbb{R})^{\mathcal{S} \times \mathcal{A}} \to [0, \infty]$ has proven particularly useful. In the univariate case, $\bar{w}_p$ is given by

$$\bar{w}_p(\nu, \nu') = \sup_{s,a \in \mathcal{S} \times \mathcal{A}} \left( \int_0^1 |F_{\nu(s,a)}^{-1}(\tau) - F_{\nu'(s,a)}^{-1}(\tau)|^p d\tau \right)^{\frac{1}{p}}, \tag{4}$$

where $p \in [1, \infty)$, $\nu, \nu'$ are any two state-action return distributions, and $F_{\nu(s,a)} : \mathbb{R} \to [0, 1]$ is the cumulative distribution function (CDF) of $\nu(s, a)$. For notational brevity, we will use the notation $w_p(\nu(s, a), \nu'(s, a)) = w_p(\nu, \nu')(s, a)$ for the $p$-Wasserstein distance between distributions $\nu, \nu'$, evaluated at $(s, a)$. One of the central insights of previous works in distributional RL is that the operator $\mathcal{T}^\pi$ is a $\gamma$-contraction in $\bar{w}_p$ (Bellemare et al., 2017), meaning that we have $\bar{w}_p(\mathcal{T}^\pi \nu, \mathcal{T}^\pi \nu') \leq \gamma \bar{w}_p(\nu, \nu')$, a property that allows us (in principle) to construct convergent value iteration schemes in the distributional setting.

### 3.2 CATEGORICAL AND QUANTILE DISTRIBUTIONAL RL

In general, we can not represent arbitrary probability distributions in $\mathscr{P}(\mathbb{R})$ and instead resort to parametric models capable of representing a subset $\mathscr{F}$ of $\mathscr{P}(\mathbb{R})$. Following Bellemare et al. (2023),

we refer to $\mathscr{F}$ as a *representation* and define it to be the set of parametric distributions $P_\theta$ with $\mathscr{F} = \{P_\theta \in \mathscr{P}(\mathbb{R}) : \theta \in \Theta\}$. Furthermore, we define the *projection operator* $\Pi : \mathscr{P}(\mathbb{R}) \to \mathscr{F}$ to be a mapping from the space of probability distributions $\mathscr{P}(\mathbb{R})$ to the representation $\mathscr{F}$. Recently, two particular choices for representation and projection have proven highly performant in deep RL: the *categorical* and *quantile* model.

The **categorical representation** (Bellemare et al., 2017; Rowland et al., 2018) assumes a weighted mixture of $K$ Dirac deltas $\delta_{z_k}$ with support at evenly spaced locations $z_k \in [z_1, ..., z_K]$. The categorical representation is then given by $\mathscr{F}_C = \{\sum_{k=1}^{K} \theta_k \delta_{z_k} | \theta_k \geq 0, \sum_{k=1}^{K} \theta_k = 1\}$. The corresponding categorical projection operator $\Pi_C$ maps a distribution $\nu$ from $\mathscr{P}(\mathbb{R})$ to a distribution in $\mathscr{F}_C$ by assigning probability mass inversely proportional to the distance to the closest $z_k$ in the support $[z_1, ..., z_K]$ for every point in the support of $\nu$. For example, for a single Dirac distribution $\delta_x$ and assuming $z_k \leq x \leq z_{k+1}$ the projection is given by

$$\Pi_C \delta_x = \frac{z_{k+1}-x}{z_{k+1}-z_k}\delta_{z_k} + \frac{x-z_k}{z_{k+1}-z_k}\delta_{z_{k+1}}. \tag{5}$$

The corner cases are defined such that $\Pi_C \delta_x = \delta_{z_1} \, \forall x \leq z_1$ and $\Pi_C \delta_x = \delta_{z_K} \, \forall x \geq z_K$. It is straightforward to extend the above projection step to finite mixtures of Dirac distributions through $\Pi_C \sum_k p_k \delta_{z_k} = \sum_k p_k \Pi_C \delta_{z_k}$. The full definition of the projection $\Pi_C$ is deferred to Appendix A.5.

The **quantile representation** (Dabney et al., 2018b), like the categorical representation, comprises mixture distributions of Dirac deltas $\delta_{\theta_k}(z)$, but in contrast parametrizes their locations rather than probabilities. This yields the representation $\mathscr{F}_Q = \{\sum_{k=1}^{K} \frac{1}{K}\delta_{\theta_k}(z) | \theta_k \in \mathbb{R}\}$. For some distribution $\nu \in \mathscr{P}(\mathbb{R})$, the quantile projection $\Pi_Q \nu$ is a mixture of $K$ Dirac delta distributions with the particular choice of locations that minimizes the 1-Wasserstein distance between $\nu \in \mathscr{P}(\mathbb{R})$ and the projection $\Pi_Q \nu \in \mathscr{F}_Q$. The parametrization $\theta_k$ with minimal 1-Wasserstein distance is given by the evaluation of the inverse of the CDF, $F_\nu^{-1}$, at midpoint quantiles $\tau_k = \frac{2k-1}{2K}$, $k \in [1, ..., K]$, s.t. $\theta_k = F_\nu^{-1}(\frac{2k-1}{2K})$. Equivalently, $\theta_k$ is the minimizer of the *quantile regression loss* (QR) (Koenker and Hallock, 2001), which is more amenable to gradient-based optimization. The loss is given by

$$\mathcal{L}_Q(\theta_k, \nu) = \mathbb{E}_{Z \sim \nu}[\rho_{\tau_k}(Z - \theta_k)], \tag{6}$$

where $\rho_\tau(u) = u(\tau - \mathbb{1}_{\{u \leq 0\}}(u))$ is an error function that assigns asymmetric weight to over- or underestimation errors and $\mathbb{1}$ denotes the indicator function.

# 4 EXPLORATION WITH DISTRIBUTIONAL PROJECTION ENSEMBLES

This paper is foremost concerned with leveraging ensembles with diverse generalization behaviors induced by different representations and projection operators. To introduce the concept of distributional projection ensembles and their properties, we describe the main components in a formal setting that foregoes sample-based stochastic approximation and function approximation, before moving to a more practical deep RL setting in Section 5. We begin by outlining the *projection mixture operator* and its contraction properties. While this does not inform an exploration algorithm in its own right, it lays a solid algorithmic foundation for the subsequently derived exploration framework. Consider an ensemble $E = \{\eta_i(s,a) \, | \, i \in [1, ..., M]\}$ of $M$ member distributions $\eta_i(s,a)$, each associated with a representation $\mathscr{F}_i$ and a projection operator $\Pi_i$. In this setting, we assume that each member distribution $\eta_i(s,a) \in \mathscr{F}_i$ is an element of the associated representation $\mathscr{F}_i$ and the projection operator $\Pi_i : \mathscr{P}(\mathbb{R}) \to \mathscr{F}_i$ maps any distribution $\nu \in \mathscr{P}(\mathbb{R})$ to $\mathscr{F}_i$ such that $\Pi_i \nu \in \mathscr{F}_i$. The set of representable uniform mixture distributions over $E$ is then given by $\mathscr{F}_E = \{\eta_E(s,a) \, | \, \eta_E(s,a) = \frac{1}{M}\sum_i \eta_i(s,a), \eta_i(s,a) \in \mathscr{F}_i, i \in [1, ..., M]\}$. We can now define a central object in this paper, the *projection mixture operator* $\Omega_M : \mathscr{P}(\mathbb{R}) \to \mathscr{F}_E$, as follows:

$$\Omega_M \eta(s,a) = \frac{1}{M}\sum_{i=1}^{M} \Pi_i \eta(s,a). \tag{7}$$

Joining $\Omega_M$ with the distributional Bellman operator $\mathcal{T}^\pi$ yields the combined operator $\Omega_M \mathcal{T}^\pi$. Fig. 2 illustrates the intuition behind the operator $\Omega_M \mathcal{T}^\pi$: the distributional Bellman operator $\mathcal{T}^\pi$ is applied to a return distribution $\eta$ (Fig. 2 a and b), then projects the resulting distribution with the individual projection operators $\Pi_i$ onto $M$ different representations $\eta_i = \Pi_i \mathcal{T}^\pi \eta \in \mathscr{F}_i$ (Fig. 2 c and d), and finally recombines the ensemble members into a mixture model in $\mathscr{F}_E$ (Fig. 2 e). In connection with iterative algorithms, we are often interested in the contractivity of the combined operator $\Omega_M \mathcal{T}^\pi$ to establish convergence. Proposition 1 delineates conditions under which we can combine individual projections $\Pi_i$ such that the resulting combined operator $\Omega_M \mathcal{T}^\pi$ is a contraction mapping.

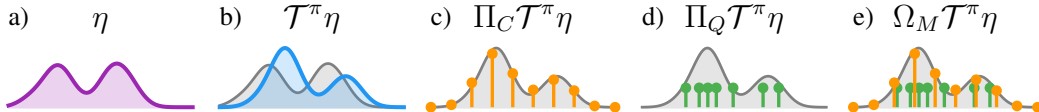

a) $\eta$  b) $\mathcal{T}^\pi \eta$  c) $\Pi_C \mathcal{T}^\pi \eta$  d) $\Pi_Q \mathcal{T}^\pi \eta$  e) $\Omega_M \mathcal{T}^\pi \eta$

Figure 2: Illustration of the projection mixture operator with quantile and categorical projections.

**Proposition 1** *Let $\Pi_i$, $i \in [1, ..., M]$ be projection operators $\Pi_i : \mathscr{P}(\mathbb{R}) \to \mathscr{F}_i$ mapping from the space of probability distributions $\mathscr{P}(\mathbb{R})$ to representations $\mathscr{F}_i$ and denote the projection mixture operator $\Omega_M : \mathscr{P}(\mathbb{R}) \to \mathscr{F}_E$ as defined in Eq. 7. Furthermore, assume that for some $p \in [1, \infty)$ each projection $\Pi_i$ is bounded in the $p$-Wasserstein metric in the sense that for any two return distributions $\eta, \eta'$ we have $w_p\big(\Pi_i \eta, \Pi_i \eta'\big)(s, a) \leq c_i w_p\big(\eta, \eta'\big)(s, a)$ for a constant $c_i$. Then, the combined operator $\Omega_M \mathcal{T}^\pi$ is bounded in the supremum $p$-Wasserstein distance $\bar{w}_p$ by*

$$\bar{w}_p(\Omega_M \mathcal{T}^\pi \eta, \Omega_M \mathcal{T}^\pi \eta') \leq \bar{c}_p \gamma \bar{w}_p(\eta, \eta')$$

*and is accordingly a contraction so long as $\bar{c}_p \gamma < 1$, where $\bar{c}_p = (\sum_{i=1}^M \frac{1}{M} c_i^p)^{1/p}$.*

The proof is deferred to Appendix A. The contraction condition in Proposition 1 is naturally satisfied for example if all projections $\Pi_i$ are non-expansions in a joint metric $w_p$. It is, however, more permissive in the sense that it only requires the joint modulus $\bar{c}_p$ to be limited, allowing for expanding operators in the ensemble for finite $p$. A contracting combined operator $\Omega_M \mathcal{T}^\pi$ allows us to formulate a simple convergent iteration scheme where in a sequence of steps $k$, ensemble members are moved toward the projected mixture distribution according to $\hat{\eta}_{i,k+1} = \Pi_i \mathcal{T}^\pi \hat{\eta}_{E,k}$, yielding the $(k+1)$-th mixture distribution $\hat{\eta}_{E,k+1} = \frac{1}{M} \sum_{i=1}^M \hat{\eta}_{i,k+1}$. This procedure can be compactly expressed by

$$\hat{\eta}_{E,k+1} = \Omega_M \mathcal{T}^\pi \hat{\eta}_{E,k}, \quad \text{for} \quad k = [0, 1, 2, 3, ...] \tag{8}$$

and has a unique fixed point which we denote $\eta_E^\pi = \hat{\eta}_{E,\infty}$.

### 4.1 FROM DISTRIBUTIONAL APPROXIMATIONS TO OPTIMISTIC BOUNDS

We proceed to describe how distributional projection ensembles can be leveraged for exploration. Our setting considers exploration strategies based on the upper-confidence-bound (UCB) algorithm (Auer, 2002). In the context of model-free RL, provably efficient algorithms often rely on the construction of a bound, that overestimates the true state-action value with high probability (Jin et al., 2018; 2020). In other words, we are interested in finding an optimistic value $\hat{Q}^+(s, a)$ such that $\hat{Q}^+(s, a) \geq Q^\pi(s, a)$ with high probability. To this end, Proposition 2 relates an estimate $\hat{Q}(s, a)$ to the true value $Q^\pi(s, a)$ through a distributional error term.

**Proposition 2** *Let $\hat{Q}(s, a) = \mathbb{E}[\hat{Z}(s, a)]$ be a state-action value estimate where $\hat{Z}(s, a) \sim \hat{\eta}(s, a)$ is a random variable distributed according to an estimate $\hat{\eta}(s, a)$ of the true state-action return distribution $\eta^\pi(s, a)$. Further, denote $Q^\pi(s, a) = \mathbb{E}[Z^\pi(s, a)]$ the true state-action, where $Z^\pi(s, a) \sim \eta^\pi(s, a)$. We have that $Q^\pi(s, a)$ is bounded from above by*

$$\hat{Q}(s, a) + w_1\big(\hat{\eta}, \eta^\pi\big)(s, a) \geq Q^\pi(s, a) \quad \forall (s, a) \in \mathcal{S} \times \mathcal{A},$$

*where $w_1$ is the 1-Wasserstein distance metric.*

The proof follows from the definition of the Wasserstein distance and is given in Appendix A. Proposition 2 implies that, for a given distributional estimate $\hat{\eta}(s, a)$, we can construct an optimistic upper bound on $Q^\pi(s, a)$ by adding a bonus of the 1-Wasserstein distance between an estimate $\hat{\eta}(s, a)$ and the true return distribution $\eta^\pi(s, a)$, which we define as $b^\pi(s, a) = w_1(\hat{\eta}, \eta^\pi)(s, a)$ in the following. By adopting an optimistic action-selection with this guaranteed upper bound on $Q^\pi(s, a)$ according to

$$a = \arg\max_{a \in \mathcal{A}} [\hat{Q}(s, a) + b^\pi(s, a)], \tag{9}$$

we maintain that the resulting policy inherits efficient exploration properties of known optimism-based exploration methods. Note that in a convergent iteration scheme, we should expect the bonus $b^\pi(s, a)$ to almost vanish in the limit of infinite iterations. We thus refer to $b^\pi(s, a)$ as a measure of the epistemic uncertainty of the estimate $\hat{\eta}(s, a)$.

### 4.2 PROPAGATION OF EPISTEMIC UNCERTAINTY THROUGH DISTRIBUTIONAL ERRORS

By Proposition 2, an optimistic policy for efficient exploration can be derived from the distributional error $b^\pi(s, a)$. However, since we do not assume knowledge of the true return distribution $\eta^\pi(s, a)$,

this error term requires estimation. The primary purpose of this section is to establish such an estimator by propagating distributional TD errors. This is necessary because the use of TD backups prohibits a consistent uncertainty quantification in values (described extensively in the Bayesian setting for example by Fellows et al. 2021). The issue is particularly easy to see by considering the backup in a single $(s, a)$ tuple: even if every estimate $\hat{\eta}_i(s, a)$ in an ensemble fits the backup $\mathcal{T}^\pi \hat{\eta}_E(s, a)$ accurately, this does not imply $\hat{\eta}_i(s, a) = \eta^\pi(s, a)$ as the TD backup may have been incorrect. Even a well-behaved ensemble (in the sense that its disagreement reliably measures prediction errors) in this case quantifies errors w.r.t. the bootstrapped target $\Omega_M \mathcal{T}^\pi \hat{\eta}_E(s, a)$, rather than the true return distribution $\eta^\pi(s, a)$.

To establish a bonus estimate that allows for optimistic action selection in the spirit of Proposition 2, we now derive a propagation scheme for epistemic uncertainty in the distributional setting. More specifically, we find that an upper bound on the bonus $b^\pi(s, a)$ satisfies a temporal consistency condition, similar to the Bellman equations, that relates the total distributional error $w_1(\hat{\eta}, \eta_E^\pi)(s, a)$ to a *one-step* error $w_1(\hat{\eta}, \Omega_M \mathcal{T}^\pi \hat{\eta})(s, a)$ that is more amenable to estimation.

**Theorem 3** *Let $\hat{\eta}(s, a) \in \mathscr{P}(\mathbb{R})$ be an estimate of the true return distribution $\eta^\pi(s, a) \in \mathscr{P}(\mathbb{R})$, and denote the projection mixture operator $\Omega_M : \mathscr{P}(\mathbb{R}) \to \mathscr{F}_E$ with members $\Pi_i$ and bounding moduli $c_i$ and $\bar{c}_p$ as defined in Proposition 1. Furthermore, assume $\Omega_M \mathcal{T}^\pi$ is a contraction mapping with fixed point $\eta_E^\pi$. We then have for all $(s, a) \in \mathcal{S} \times \mathcal{A}$*

$$w_1(\hat{\eta}, \eta_E^\pi)(s, a) \leq w_1(\hat{\eta}, \Omega_M \mathcal{T}^\pi \hat{\eta})(s, a) + \bar{c}_1 \gamma \mathbb{E}\left[w_1(\hat{\eta}, \eta_E^\pi)(S_1, A_1) \big| S_0 = s, A_0 = a\right],$$

*where $S_1 \sim P(\cdot | S_0 = s, A_0 = a)$ and $A_1 \sim \pi(\cdot | S_1)$.*

The proof is given in Appendix A and exploits the triangle inequality property of the Wasserstein distance. It may be worth noting that Theorem 3 is a general result that is not restricted to the use of projection ensembles. It is, however, a natural complement to the iteration described in Eq. 8 in that it allows us to reconcile the benefits of bootstrapping diverse ensemble mixtures with optimistic action selection for directed exploration. To this end, we devise a separate iteration procedure aimed at finding an approximate upper bound on $w_1(\hat{\eta}, \eta_E^\pi)(s, a)$. Denoting the $k$-th iterate of the bonus estimate $\hat{b}_k(s, a)$, we have by Theorem 3 that the iteration

$$\hat{b}_{k+1}(s, a) = w_1(\hat{\eta}, \Omega_M \mathcal{T}^\pi \hat{\eta})(s, a) + \bar{c}_1 \gamma \mathbb{E}_{P, \pi}\left[\hat{b}_k(S_1, A_1) \big| S_0 = s, A_0 = a\right] \ \forall (s, a) \in \mathcal{S} \times \mathcal{A},$$

converges to an upper bound on $w_1(\hat{\eta}, \eta_E^\pi)(s, a)$[1]. Notably, this iteration requires only a local error estimate $w_1(\hat{\eta}, \Omega_M \mathcal{T}^\pi \hat{\eta})(s, a)$ and is more amenable to estimation through our ensemble.

We conclude this section with the remark that the use of projection ensembles may clash with the intuition that epistemic uncertainty should vanish in convergence. This is because each member inherits irreducible approximation errors from the projections $\Pi_i$. In Appendix A, we provide general bounds for these errors and show that residual errors can be controlled through the number of atoms $K$ in the specific example of an ensemble based on the quantile and categorical projections.

## 5 Deep distributional RL with projection ensembles

Section 4 has introduced the concept of projection ensembles in a formal setting. In this section, we aim to transcribe the previously derived algorithmic components into a deep RL algorithm that departs from several of the previous assumptions. Specifically, this includes 1) control with a greedy policy, 2) sample-based stochastic approximation, 3) nonlinear function approximation, and 4) gradient-based optimization. While this sets the following section apart from the theoretical setting considered in Section 4, we hypothesize that diverse projection ensembles bring to bear several advantages in this scenario. The underlying idea is that distributional projections and the functional constraints they entail offer an effective tool to impose diverse generalization behaviors on an ensemble, yielding a more reliable tool for out-of-distribution sample detection. In particular, we implement the above-described algorithm with a neural ensemble comprising the models of the two popular deep RL algorithms quantile regression deep Q network (QR-DQN) (Dabney et al., 2018b) and C51 (Bellemare et al., 2017).

---

[1]To see the convergence, note that the sequence is equivalent to an iteration with $T^\pi$ in an MDP with the deterministic immediate reward $w_1(\hat{\eta}, \Omega_M \mathcal{T}^\pi \hat{\eta})(s, a)$.

## 5.1 DEEP QUANTILE AND CATEGORICAL PROJECTION ENSEMBLES FOR EXPLORATION

In this section, we propose Projection Ensemble DQN (PE-DQN), a deep RL algorithm that combines the quantile and categorical projections (Dabney et al., 2018b; Bellemare et al., 2017) into a diverse ensemble to drive exploration and learning stability. Our parametric model consists of the mixture distribution $\eta_{E,\theta}$ parametrized by $\theta$. We construct $\eta_{E,\theta}$ as an equal mixture between a quantile and a categorical representation, each parametrized through a NN with $K$ output logits where we use the notation $\theta_{ik}$ to mean the $k$-th logit of the network parametrized by the parameters $\theta_i$ of the $i$-th model in the ensemble. We consider a sample transition $(s, a, r, s', a')$ where $a'$ is chosen greedily according to $\mathbb{E}_{Z \sim \eta_{E,\theta}(s',a')}[Z]$. Dependencies on $(s, a)$ are hereafter dropped for conciseness by writing $\theta_{ik} = \theta_{ik}(s, a)$ and $\theta'_{ik} = \theta_{ik}(s', a')$.

**Projection losses.** Next, we assume that bootstrapped return distributions are generated by a set of delayed parameters $\tilde{\theta}$, as is common (Mnih et al., 2015). The stochastic (sampled) version of the distributional Bellman operator $\hat{\mathcal{T}}^\pi$, applied to the target ensemble's mixture distribution $\eta_{E,\tilde{\theta}}$ yields

$$\hat{\mathcal{T}}^\pi \eta_{E,\tilde{\theta}} = \tfrac{1}{2} \sum_{i=1}^{M=2} \sum_{k=1}^{K} p(\tilde{\theta}'_{ik}) \, \delta_{r+\gamma z(\tilde{\theta}'_{ik})}. \tag{10}$$

Instead of applying the projection mixture $\Omega_M$ analytically, as done in Section 4, the parametric estimates $\eta_{E,\theta}$ are moved incrementally towards a projected target distribution through gradient descent on a loss function. In the *quantile* representation, we augment the classical quantile regression loss (Koenker and Hallock, 2001) with an importance-sampling ratio $Kp(\tilde{\theta}'_{ij})$ to correct for the non-uniformity of atoms from the bootstrapped distribution $\hat{\mathcal{T}}^\pi \eta_{E,\tilde{\theta}}$. For a set of fixed quantiles $\tau_k$, the loss $\mathcal{L}_1$ is given by

$$\mathcal{L}_1\big(\eta_{\theta_1}, \Pi_Q \hat{\mathcal{T}}^\pi \eta_{E,\tilde{\theta}}\big) = \sum_{i=1}^{M=2} \sum_{k,j=1}^{K} Kp(\tilde{\theta}'_{ij})\Big(\rho_{\tau_k}\big(r + \gamma z(\tilde{\theta}'_{ij}) - \theta_{1k}\big)\Big). \tag{11}$$

The *categorical* model minimizes the Kullback-Leibler (KL) divergence between the projected bootstrap distribution $\Pi_C \hat{\mathcal{T}}^\pi \eta_{E,\tilde{\theta}}$ and an estimate $\eta_{\theta_2}$. The corresponding loss is given by

$$\mathcal{L}_2\big(\eta_{\theta_2}, \Pi_C \hat{\mathcal{T}}^\pi \eta_{E,\tilde{\theta}}\big) = D_{KL}\big(\Pi_C \hat{\mathcal{T}}^\pi \eta_{E,\tilde{\theta}} \| \eta_{\theta_2}\big). \tag{12}$$

As $\hat{\mathcal{T}}^\pi \eta_{E,\tilde{\theta}}$ is a mixture of Dirac distributions, the definition of the projection $\Pi_C$ according to Eq. 5 can be applied straightforwardly to obtain the projected bootstrap distribution $\Pi_C \hat{\mathcal{T}}^\pi \eta_{E,\tilde{\theta}}$.

**Uncertainty Propagation.** We aim to estimate a state-action dependent bonus $b_\phi(s, a)$ in the spirit of Theorem 3 and the subsequently derived iteration with a set of parameters $\phi$. For this, we estimate the local error estimate $w_1(\eta_{E,\theta}, \Omega_M \hat{\mathcal{T}}^\pi \eta_{E,\theta})(s, a)$ as the average ensemble disagreement $w_{\mathrm{avg}}(s, a) = 1/(M(M-1)) \sum_{i,j=1}^{M} w_1(\eta_{\theta_i}, \eta_{\theta_j})(s, a)$. The bonus $b_\phi(s, a)$ can then be learned in the same fashion as a regular value function with the local uncertainty estimate $w_{\mathrm{avg}}(s, a)$ as an intrinsic reward. This yields the exploratory action-selection rule

$$a_\epsilon = \arg\max_{a \in \mathcal{A}} \big(\mathbb{E}_{Z \sim \eta_{E,\theta}(s,a)}[Z] + \beta \, b_\phi(s, a)\big), \tag{13}$$

where $\beta$ is a hyperparameter to control the policy's drive towards exploratory actions. Further details on our implementation and an illustration of the difference between local error estimates and bonus estimates in practice are given in Appendix B.2 and Appendix B.3.

## 6 EXPERIMENTAL RESULTS

Our experiments are designed to provide us with a better understanding of how PE-DQN operates, in comparison to related algorithms as well as in relation to its algorithmic elements. To this end, we aimed to keep codebases and hyperparameters between all implementations equal up to algorithm-specific parameters, which we optimized with a grid search on a selected subsets of problems. Further details regarding the experimental design and implementations are provided in Appendix B.

We outline our choice of baselines briefly: Bootstrapped DQN with prior functions (BDQN+P) (Osband et al., 2019) approximates posterior sampling of a parametric value function by combining statistical bootstrapping with additive prior functions in an ensemble of DQN agents. Information-directed sampling (IDS-C51) (Nikolov et al., 2019) builds on the BDQN+P architecture but acts

according to an information-gain ratio for which Nikolov et al. (2019) estimate aleatoric uncertainty (noise) with the categorical C51 model. In contrast, Decaying Left-Truncated Variance (DLTV) QR-DQN (Mavrin et al., 2019) uses a distributional value approximation based on the quantile representation and follows a decaying exploration bonus of the left-truncated variance.

## 6.1 Do different projections lead to different generalization behavior?

First, we examine empirically the influence of the projection step in deep distributional RL on generalization behaviors. For this, we probe the influence of the quantile and categorical projections on generalization through an experiment that evaluates exploration in a reward-free setting. Specifically, we equip agents with an action-selection rule that maximizes a particular statistic $\mathbb{S}[Z]$ of the predictive distribution $\hat{\eta}(s, a)$ according to

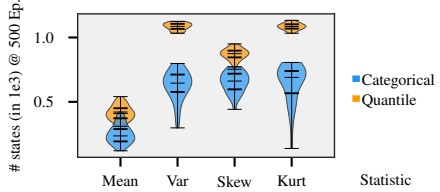

Figure 3: Deep-sea exploration with different statistics. Higher means more exploration. Bars represent medians and interquartile ranges of 30 seeds.

$$a = \arg\max_{a \in \mathcal{A}} \left( \mathbb{S}[Z] \right), Z \sim \hat{\eta}(s, a).$$

The underlying idea is that this selection rule leads to exploration of novel state-action regions only if high values of the statistic are correlated with high epistemic uncertainty. For example, if we choose a quantile representation with $\mathbb{S}[Z]$ to be the variance of the distribution, we recover a basic form of the exploration algorithm DLTV-QR (Mavrin et al., 2019). Fig. 3 shows the results of this study for the first four statistical moments on the deep exploration benchmark *deep sea* with size 50. Except for the mean (the greedy policy), the choice of projection influences significantly whether the statistic-maximizing policy leads to more exploration, implying that the generalization behaviour of the 2nd to 4th moment of the predictive distributions is shaped distinctly by the employed projection.

## 6.2 The behaviour suite

In order to assess the learning process of agents in various aspects on a wide range of tasks, we evaluate PE-DQN on the behavior suite (bsuite) (Osband et al., 2020), a battery of benchmark problems constructed to assess key properties of RL algorithms. The suite consists of 23 tasks with up to 22 variations in size or seed, totaling 468 environments.

**Comparative evaluation.** Fig. 4 (a) shows the results of the entire bsuite experiment, summarized in seven *core capabilities*. These capability scores are computed as proposed by Osband et al. (2020) and follow a handcrafted scoring function per environment. For example, exploration capability is scored by the average regret in the sparse-reward environments *deep sea*, *stochastic deep sea*, and *cartpole swingup*. The full set of results is provided in Appendix B. Perhaps unsurprisingly, PE-DQN has its strongest performance in the exploration category but we find that it improves upon baselines in several more categories. Note here that PE-DQN uses substantially fewer models than the baselines, with a total of 4 distributional models compared to the 20 DQN models used in the ensembles of both BDQN+P and IDS, where the latter requires an additional C51 model.

## 6.3 The deep-sea environment and ablations

*Deep sea* is a hard exploration problem in the behavior suite and has recently gained popularity as an exploration benchmark (Osband et al., 2019; Janz et al., 2019; Flennerhag et al., 2020). It is a sparse reward environment where agents can reach the only rewarding state at the bottom right of an $N \times N$ grid through a unique sequence of actions in an exponentially growing trajectory space. We ran an additional experiment on deep sea with grid sizes up to 100; double the maximal size in the behavior suite. Fig. 4 (b) shows a summary of this experiment where we evaluated episodic regret, that is the number of non-rewarding episodes with a maximum budget of 10000 episodes. PE-DQN scales more gracefully to larger sizes of the problem than the baselines, reducing the median regret by roughly half. The r.h.s. plot in Fig. 4 (b) shows the results of ablation studies designed to provide a more nuanced view of PE-DQN's performance; the baselines labeled PE-DQN[QR/QR] and PE-DQN[C51/C51] use the same bonus estimation step as PE-DQN except that ensemble members consist of equivalent models with *the same projections and representations*. Conversely, PE-DQN [Ind.] uses PE-DQN's diverse projection ensemble and employs an optimistic action-selection directly with the ensemble disagreement $w_{\mathrm{avg}}(s, a)$ but trains models independently and accordingly does not make use of an

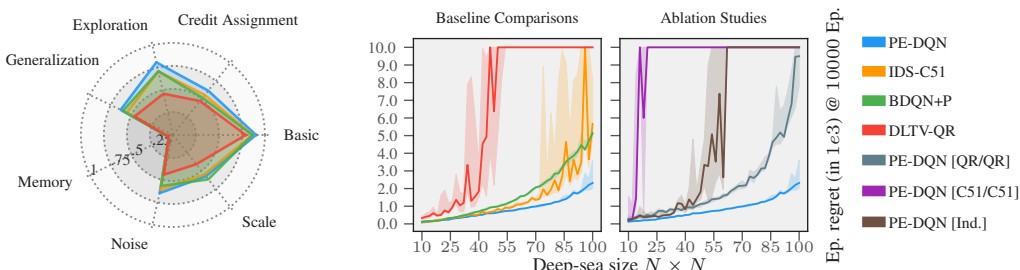

Figure 4: (a) Summary of bsuite experiments. Wide is better. (b) Median episodic regret for deep sea sizes up to 100. Low is better. Shaded regions are the interquartile range of 10 seeds.

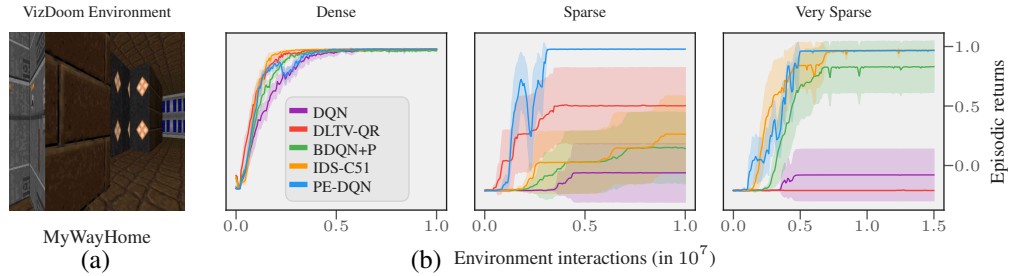

Figure 5: (a) Visual observation in the VizDoom environment (Kempka et al., 2016). (b) Mean learning curves in different variations of the *MyWayHome* VizDoom environment. Shaded regions are 90% Student's t confidence intervals from 10 seeds.

uncertainty propagation scheme in the spirit of Theorem 3. Both components lead to a pronounced difference in exploration capability and rendered indispensable to PE-DQN's overall performance.

## 6.4 THE VIZDOOM ENVIRONMENT

We investigate PE-DQN's behavior in a high-dimensional visual domain. The *VizDoom* environment *MyWayHome* (Kempka et al., 2016) tasks agents with finding a (rewarding) object by navigating in a maze-like map with ego-perspective pixel observations as seen in Fig. 5 (a). Following work by Pathak et al. (2017), we run three variations of this experiment where the reward sparsity is increased by spawning the player further away from the goal object. Learning curves for all algorithms are shown in Fig. 5 (b). Among the tested algorithms, only PE-DQN finds the object across all 10 seeds in all environments, indicating particularly reliable novelty detection. Interestingly, the sparse domain proved harder to baseline algorithms which we attribute to the "forkedness" of the associated map (see Appendix B). This result moreover shows that diverse projection ensembles scale gracefully to high-dimensional domains while using significantly fewer models than the ensemble-based baselines.

## 7 CONCLUSION

In this work, we have introduced projection ensembles for distributional RL, a method combining models based on different parametric representations and projections of return distributions. We provided a theoretical analysis that establishes convergence conditions and bounds on residual approximation errors that apply to general compositions of such projection ensembles. Furthermore, we introduced a general propagation method that reconciles one-step distributional TD errors with optimism-based exploration. PE-DQN, a deep RL algorithm, empirically demonstrates the efficacy of diverse projection ensembles on exploration tasks and showed performance improvements on a wide range of tasks. We believe our work opens up a number of promising avenues for future research. For example, we have only considered the use of uniform mixtures over distributional ensembles in this work. A continuation of this approach may aim to use a diverse collection of models less conservatively, aiming to exploit the strengths of particular models in specific regions of the state-action space.

## 8 ACKNOWLEDGEMENTS

We thank Max Weltevrede, Pascal van der Vaart, Miguel Suau, and Yaniv Oren for fruitful discussions and remarks. The project has received funding from the EU Horizon 2020 programme under grant number 964505 (Epistemic AI). The computational resources for empirical work were provided by the Delft High Performance Computing Centre (DHPC) and the Delft Artificial Intelligence Cluster (DAIC).

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

# A  APPENDIX

This section provides proofs for the theoretical claims and establishes further results on the residual approximation error incurred by our method.

## A.1  PROOF OF PROPOSITION 1

Before stating supporting lemmas and proofs of the results in Section 4, we recall several basic properties of the $p$-Wasserstein distances which we will find useful in the subsequent proofs. Derivations of these properties can for example be found in an overview by Mariucci and Reiß (2018) .

**P.1** The $p$-Wasserstein distances satisfy the *triangle inequality*, that is
$$w_p(X,Y) \leq w_p(X,Z) + w_p(Z,Y) \,.$$

**P.2** For random variables $X$ and $Y$ and an auxiliary variable $Z$ independent of $X$ and $Y$, the $p$-Wasserstein metric satisfies the inequality
$$w_p(X+Z, Y+Z) \leq w_p(X,Y) \,.$$

**P.3** For a real-valued scalar $a \in \mathbb{R}$, we have
$$w_p(aX, aY) = |a| w_p(X,Y) \,.$$

**Lemma 4** *Let* $\nu = \sum_{i=1}^{M} \frac{1}{M} \nu_i$, $\nu' = \sum_{i=1}^{M} \frac{1}{M} \nu'_i$ *be two mixture distributions* $\nu, \nu' \in \mathscr{P}(\mathbb{R})$. *Furthermore denote* $w_p(\nu, \nu')$ *the p-Wasserstein metric between* $\nu$ *and* $\nu'$. *Then* $w_p^p$ *satisfies*
$$w_p^p(\nu, \nu') \leq \frac{1}{M} \sum_{i=1}^{M} w_p^p(\nu_i, \nu'_i).$$

*Proof.* The Wasserstein distance in its general form is expressed in terms of couplings between the probability measures $\nu$ and $\nu'$ according to
$$w_p(\nu, \nu') = \inf_{\mu \in \Gamma(\nu, \nu')} \mathbb{E}_{(x,y) \sim \mu}[|x-y|^p]^{1/p},$$

where $\Gamma(\nu, \nu')$ is the set of all couplings between $\nu$ and $\nu'$, i.e. joint distributions on $\mathscr{P}(\mathbb{R}^2)$ with marginals $\nu$ and $\nu'$. Now suppose for each $i$ we have a coupling $\mu_i(x,y) \in \Gamma(\nu_i, \nu'_i)$ such that
$$\mathbb{E}_{(x,y) \sim \mu_i}[|x-y|^p] = \inf_{\mu \in \Gamma(\nu_i, \nu'_i)} \mathbb{E}_{(x,y) \sim \mu}[|x-y|^p] = w_p^p(\nu_i, \nu'_i).$$

Since by definition $\mu_i(x,y)$ is a coupling of $\nu_i$ and $\nu'_i$, the mixture of couplings $\bar{\mu}(x,y) = \sum_{i=1}^{M} \frac{1}{M} \mu_i(x,y)$ is then a valid coupling of $\nu$ and $\nu'$, as $\int \bar{\mu}(x,y)dy = \nu(x)$ and $\int \bar{\mu}(x',y)dx' = \nu'(x)$. We can thus write
$$
\begin{aligned}
w_p^p(\nu, \nu') &= \inf_{\mu \in \Gamma(\nu, \nu')} \mathbb{E}_{(x,y) \sim \mu}[|x-y|^p] \\
&\leq \mathbb{E}_{(x,y) \sim \bar{\mu}}[|x-y|^p] \\
&= \sum_{i=1}^{M} \frac{1}{M} \mathbb{E}_{(x,y) \sim \mu_i}[|x-y|^p] \\
&= \sum_{i=1}^{M} \frac{1}{M} w_p^p(\nu_i, \nu'_i) \,.
\end{aligned}
$$

**Proposition 1** *Let* $\Pi_i$, $i \in [1, ..., M]$ *be projection operators* $\Pi_i : \mathscr{P}(\mathbb{R}) \to \mathscr{F}_i$ *mapping from the space of probability distributions* $\mathscr{P}(\mathbb{R})$ *to representations* $\mathscr{F}_i$ *and denote the projection mixture operator* $\Omega_M : \mathscr{P}(\mathbb{R}) \to \mathscr{F}_E$ *as defined in Eq. 7. Furthermore, assume that for some* $p \in [1, \infty)$ *each projection* $\Pi_i$ *is bounded in the $p$-Wasserstein metric in the sense that for any two return distributions* $\eta, \eta'$ *we have* $w_p\big(\Pi_i \eta, \Pi_i \eta'\big)(s,a) \leq c_i w_p\big(\eta, \eta'\big)(s,a)$ *for a constant* $c_i$. *Then, the combined operator* $\Omega_M \mathcal{T}^\pi$ *is bounded in the supremum $p$-Wasserstein distance* $\bar{w}_p$ *by*
$$\bar{w}_p(\Omega_M \mathcal{T}^\pi \eta, \Omega_M \mathcal{T}^\pi \eta') \leq \bar{c}_p \gamma \bar{w}_p(\eta, \eta')$$
*and is accordingly a contraction so long as* $\bar{c}_p \gamma < 1$, *where* $\bar{c}_p = (\sum_{i=1}^{M} \frac{1}{M} c_i^p)^{1/p}$.

*Proof.* Due to the assumption of the proposition, we have $w_p(\Pi_i \nu, \Pi_i \nu') \leq c_i w_p(\nu, \nu')$. With Lemma 4 and the $\gamma$-contractivity of $\mathcal{T}^\pi$, it follows that

$$\bar{w}_p^p(\Omega_M \mathcal{T}^\pi \eta, \Omega_M \mathcal{T}^\pi \eta') = \bar{w}_p^p(\sum_{i=1}^M \tfrac{1}{M} \Pi_i \mathcal{T}^\pi \eta, \sum_{i=1}^M \tfrac{1}{M} \Pi_i \mathcal{T}^\pi \eta')$$

$$\leq \tfrac{1}{M} \sum_{i=1}^M \bar{w}_p^p(\Pi_i \mathcal{T}^\pi \eta, \Pi_i \mathcal{T}^\pi \eta')$$

$$\leq \tfrac{1}{M} \sum_{i=1}^M c_i^p \bar{w}_p^p(\mathcal{T}^\pi \eta, \mathcal{T}^\pi \eta')$$

$$\leq \tfrac{1}{M} \sum_{i=1}^M c_i^p \gamma^p \bar{w}_p^p(\eta, \eta')$$

$$= \gamma^p \bar{w}_p^p(\eta, \eta') \tfrac{1}{M} \sum_{i=1}^M c_i^p.$$

The state then finally follows by taking the $p$-th root, yielding the joint modulus $\bar{c}_p = (\sum_{i=1}^M \tfrac{1}{M} c_i^p)^{1/p}$.

## A.2 PROOF OF PROPOSITION 2

**Proposition 2** *Let $\hat{Q}(s, a) = \mathbb{E}[\hat{Z}(s, a)]$ be a state-action value estimate where $\hat{Z}(s, a) \sim \hat{\eta}(s, a)$ is a random variable distributed according to an estimate $\hat{\eta}(s, a)$ of the true state-action return distribution $\eta^\pi(s, a)$. Further, denote $Q^\pi(s, a) = \mathbb{E}[Z^\pi(s, a)]$ the true state-action, where $Z^\pi(s, a) \sim \eta^\pi(s, a)$. We have that $Q^\pi(s, a)$ is bounded from above by*

$$\hat{Q}(s, a) + w_1(\hat{\eta}, \eta^\pi)(s, a) \geq Q^\pi(s, a) \quad \forall (s, a) \in \mathcal{S} \times \mathcal{A},$$

*where $w_1$ is the 1-Wasserstein distance metric.*

*Proof.* We begin by stating a property that relates the expected value $\mathbb{E}[X]$ to the CDF of $X$ under the condition that the expectation $\mathbb{E}[X]$ is well-defined and finite. The property is an extension to the property of the expectation of nonnegative variables which itself is a consequence of Fubini's Theorem (see for example Ibe 2014 for this). Let $X \sim \nu$ and write $F_\nu$ for the CDF of $\nu$, then:

$$\mathbb{E}[X] = \int_0^\infty \big(1 - F_\nu(x)\big) dx - \int_{-\infty}^0 F_\nu(x) dx.$$

Now, suppose an auxiliary variable $X'$ is distributed according to the law $\nu'$. It then follows that

$$\big|\mathbb{E}[X] - \mathbb{E}[X']\big| = \Big| \int_0^\infty \big(F_{\nu'}(x) - F_\nu(x)\big) dx - \int_{-\infty}^0 \big(F_\nu - F_{\nu'}(x)\big) dx \Big|$$

$$= \Big| \int_{-\infty}^\infty F_{\nu'}(x) - F_\nu(x) dx \Big|$$

$$\leq \int_{-\infty}^\infty \big|F_{\nu'}(x) - F_\nu(x)\big| dx$$

$$= w_1(\nu, \nu'),$$

where the last step was obtained by a change of variables in the definition of the 1-Wasserstein distance:

$$w_1(\nu, \nu') = \int_0^1 |F_\nu^{-1}(\tau) - F_{\nu'}^{-1}(\tau)| d\tau$$

$$= \int_{\mathbb{R}} |F_\nu(x) - F_{\nu'}(x)| dx.$$

The result of Proposition 2 is obtained by rearranging.

## A.3 PROOF OF THEOREM 3

Before stating the proof of Theorem 3, we formalize the notion of a pushforward distribution which will be useful in a more explicit description of the distributional Bellman operator $\mathcal{T}^\pi$. Our notation here follows the detailed exposition by Bellemare et al. (2023) .

**Definition 5** *For a function $f : \mathbb{R} \to \mathbb{R}$ and a random variable $Z$ with distribution $\nu = \mathcal{D}(Z)$, $\nu \in \mathscr{P}(\mathbb{R})$, the pushforward distribution $f_{\#}\nu \in \mathscr{P}(\mathbb{R})$ of $\nu$ through $f$ is defined as*

$$f_{\#}\nu(B) = \nu(f^{-1}(B)), \quad \forall B \in \mathcal{B}(\mathbb{R}),$$

*where $\mathcal{B}$ are the Borel subsets of $\mathbb{R}$.*

Equivalently to Definition 5, we may write $f_{\#}\nu = \mathcal{D}(f(Z))$. By defining a bootstrap transformation $b_{r,\gamma} : \mathbb{R} \to \mathbb{R}$ with $b_{r,\gamma} = r + \gamma x$, we can state a more explicit definition of the distributional Bellman operator $\mathcal{T}^{\pi}$ according to Definition 6.

**Definition 6** *[Distributional Bellman Operator (Bellemare et al., 2017) ] The distributional Bellman operator $\mathcal{T}^{\pi} : \mathscr{P}(\mathbb{R})^{\mathcal{S} \times \mathcal{A}} \to \mathscr{P}(\mathbb{R})^{\mathcal{S} \times \mathcal{A}}$ is given by*

$$\big(\mathcal{T}^{\pi}\eta\big)(s,a) = \mathbb{E}\big[(b_{R_0,\gamma})_{\#}\eta(S_1, A_1)\big|S_0 = s, A_0 = a\big],$$

*where $S_1 \sim P(\cdot|S_0 = s, A_0 = a), \quad A_1 \sim \pi(\cdot|S_1)$.*

**Lemma 7** *Let $(b_{r,\gamma})_{\#}\nu \in \mathscr{P}(\mathbb{R})$ be the pushforward distribution of $\nu \in \mathscr{P}(\mathbb{R})$ through $b_{r,\gamma} : \mathbb{R} \to \mathbb{R}$. Then we have for two distributions $\nu, \nu'$ and the 1-Wasserstein distance $w_1$ that*

$$w_1\big((b_{r,\gamma})_{\#}\nu, (b_{r,\gamma})_{\#}\nu'\big) = \gamma w_1(\nu, \nu').$$

*Proof.* The proof follows from the definition of the 1-Wasserstein distance. Let $Z \sim \nu$ and $Z' \sim \nu'$ be two independent random variables, then

$$w_1\big((b_{r,\gamma})_{\#}\nu, (b_{r,\gamma})_{\#}\nu'\big) = w_1\big(\mathcal{D}(r + \gamma Z), \mathcal{D}(r + \gamma Z')\big)$$

$$= \int_0^1 \big|F^{-1}_{(b_{0,\gamma})_{\#}\nu}(\tau) - F^{-1}_{(b_{0,\gamma})_{\#}\nu'}(\tau)\big|d\tau$$

$$= |\gamma|w_1(\nu, \nu').$$

**Theorem 3** *Let $\hat{\eta}(s,a) \in \mathscr{P}(\mathbb{R})$ be an estimate of the true return distribution $\eta^{\pi}(s,a) \in \mathscr{P}(\mathbb{R})$, and denote the projection mixture operator $\Omega_M : \mathscr{P}(\mathbb{R}) \to \mathscr{F}_E$ with members $\Pi_i$ and bounding moduli $c_i$ and $\bar{c}_p$ as defined in Proposition 1. Furthermore, assume $\Omega_M \mathcal{T}^{\pi}$ is a contraction mapping with fixed point $\eta_E^{\pi}$. We then have for all $(s,a) \in \mathcal{S} \times \mathcal{A}$*

$$w_1\big(\hat{\eta}, \eta_E^{\pi}\big)(s,a) \leq w_1\big(\hat{\eta}, \Omega_M \mathcal{T}^{\pi}\hat{\eta}\big)(s,a) + \bar{c}_1\,\gamma\,\mathbb{E}\big[w_1\big(\hat{\eta}, \eta_E^{\pi}\big)(S_1, A_1)\big|S_0 = s, A_0 = a\big],$$

*where $S_1 \sim P(\cdot|S_0 = s, A_0 = a)$ and $A_1 \sim \pi(\cdot|S_1)$.*

*Proof.* Since $\eta_E^{\pi}(s,a)$ is the fixed point of the combined operator $\Omega_M \mathcal{T}^{\pi}$, we have that $\Omega_M \mathcal{T}^{\pi}\eta_E^{\pi}(s,a) = \eta_E^{\pi}(s,a)$. From the triangle inequality it follows that

$$w_1\big(\hat{\eta}, \eta_E^{\pi}\big)(s,a) \leq w_1\big(\hat{\eta}, \Omega_M \mathcal{T}^{\pi}\hat{\eta}\big)(s,a) + w_1\big(\Omega_M \mathcal{T}^{\pi}\hat{\eta}, \Omega_M \mathcal{T}^{\pi}\eta_E^{\pi}\big)(s,a). \tag{14}$$

Furthermore, for the second term on the r.h.s. in Eq. 14 the following holds:

$$w_1\big(\Omega_M \mathcal{T}^{\pi}\hat{\eta}, \Omega_M \mathcal{T}^{\pi}\eta_E^{\pi}\big)(s,a) = w_1\big(\tfrac{1}{M}\sum_{i=1}^M \Pi_i \mathcal{T}^{\pi}\hat{\eta}, \tfrac{1}{M}\sum_{i=1}^M \Pi_i \mathcal{T}^{\pi}\eta_E^{\pi}\big)(s,a)$$

$$\leq \tfrac{1}{M}\sum_{i=1}^M c_i w_1\big(\mathcal{T}^{\pi}\hat{\eta}, \mathcal{T}^{\pi}\eta_E^{\pi}\big)(s,a)$$

$$= \bar{c}_1 w_1\big(\mathcal{T}^{\pi}\hat{\eta}, \mathcal{T}^{\pi}\eta_E^{\pi}\big)(s,a).$$

Under slight abuse of the assumptions in Section 3, we here consider an immediate reward distribution with finite support on $\mathcal{R}$ to simplify the following derivation. In this case, we can write out the expectation in Definition 6 as

$$\big(\mathcal{T}^{\pi}\hat{\eta}\big)(s,a) = \sum_{r \in \mathcal{R}}\sum_{s' \in \mathcal{S}}\sum_{a' \in \mathcal{A}} Pr(R_0 = r, A_1 = a', S_1 = s'|S_0 = s, A_0 = a)\big((b_{r,\gamma})_{\#}\hat{\eta}(s',a')\big),$$

where $Pr(\cdot)$ is the joint probability distribution given by the transition kernel $P(\cdot|s,a)$, the immediate reward distribution $\mathcal{R}(\cdot|s,a)$, and the policy $\pi(\cdot|S')$. Thus, by Lemma 4 and Lemma 7 it follows that

$$\bar{c}_1 w_1\big(\mathcal{T}^{\pi}\hat{\eta}, \mathcal{T}^{\pi}\eta_E^{\pi}\big)(s,a)$$

$$\leq \bar{c}_1\mathbb{E}\big[w_1\big((b_{R_0,\gamma})_{\#}\hat{\eta}(S_1, A_1), (b_{R_0,\gamma})_{\#}\eta_E^{\pi}(S_1, A_1)\big)\big|S_0 = s, A_0 = a\big]$$

$$= \bar{c}_1\gamma\mathbb{E}\big[w_1\big(\hat{\eta}, \eta_E^{\pi}\big)(S_1, A_1)\big|S_0 = s, A_0 = a\big],$$

where $S_1 \sim P(\cdot|S_0 = s, A_0 = a)$ and $A_1 \sim \pi(\cdot|S')$. The proof is completed by rearranging.

A.4 RESIDUAL EPISTEMIC UNCERTAINTY

Due to a limitation to finite-dimensional representations and the use of varying projections, our algorithm incurs residual approximation errors which may not vanish even in convergence. In the context of epistemic uncertainty quantification, this is unfortunate as it can frustrate exploration or lead to overconfident predictions. Specifically, the undesired properties are twofold: 1) Even in convergence, the fixed point $\eta_E^\pi$ does not equal the true return distribution (bias). 2) Even in the fixed point $\eta_E^\pi$, the ensemble disagreement $w_{avg}$ does not vanish. Often, however, we may be able to upper bound and control the error incurred due to the projections $\Pi_i$. In this case, Propositions 8 and 9 provide upper bounds on both types of errors as a function of bounded projection errors.

**Proposition 8** *Let $\Omega_M$ be a projection mixture operator with individual projections $\Pi_i$ defined as in Eq. (7) . Further, assume each projection $\Pi_i$ is upper bounded by $w_p(\Pi_i\nu, \nu) \leq d_i$ for some $p \in [1, \infty)$. Then, the $p$-Wasserstein distance between the fixed point $\eta_E^\pi(s, a) = \Omega_M \mathcal{T}^\pi \eta_E^\pi(s, a)$ and the true return distribution $\eta^\pi(s, a) = \mathcal{T}^\pi \eta^\pi(s, a)$ satisfies*

$$w_p(\eta_E^\pi, \eta^\pi)(s, a) \; \leq \; \tfrac{\bar{d}_p}{1 - \bar{c}_p \gamma} \quad \forall (s, a) \in \mathcal{S} \times \mathcal{A}, \qquad \text{where} \qquad \bar{d}_p \; = \; (\sum_{i=1}^M \tfrac{1}{M} d_i^p)^{1/p} \,.$$

*Proof.* To show the desired property, we will make use of Proposition 1 and Lemma 4. We omitted the dependency on $(s, a)$ in this section for brevity. It follows then from the triangle inequality that

$$w_p(\eta_E^\pi, \eta^\pi) \leq w_p(\Omega_M \mathcal{T}^\pi \eta_E^\pi, \Omega_M \eta^\pi) + w_p(\Omega_M \eta^\pi, \eta^\pi)$$
$$= w_p(\Omega_M \mathcal{T}^\pi \eta_E^\pi, \Omega_M \mathcal{T}^\pi \eta^\pi) + w_p(\Omega_M \eta^\pi, \eta^\pi)$$
$$\leq \bar{c}_p \gamma w_p(\eta_E^\pi, \eta^\pi) + w_p(\tfrac{1}{M} \sum_{i=1}^M \Pi_i \eta^\pi, \eta^\pi)$$
$$\leq \bar{c}_p \gamma w_p(\eta_E^\pi, \eta^\pi) + (\tfrac{1}{M} \sum_{i=1}^M w_p^p(\Pi_i \eta^\pi, \eta^\pi))^{1/p} \,.$$

Per the assumption of Proposition 8 and by rearranging we obtain the desired result.

**Proposition 9** *Let $w_{avg}$ be the average ensemble disagreement given by $\frac{1}{M(M-1)} \sum_{i,j=1}^M w_p(\hat{\eta}_i, \hat{\eta}_j)$ and assume individual projections $\Pi_i$ are bounded by $w_p(\Pi_i\nu, \nu) \leq d_i$. For an ensemble $E$ whose mixture distribution equals exactly the fixed point $\eta_E^\pi(s, a) = \Omega_M \mathcal{T}^\pi \eta_E^\pi(s, a)$, the average ensemble disagreement $w_{avg}$ satisfies the inequality*

$$w_{avg}(s, a) \; \leq \; \tfrac{2M}{M-1} \bar{d} \quad \forall (s, a) \in \mathcal{S} \times \mathcal{A}, \qquad \text{where} \qquad \bar{d} \; = \; \tfrac{1}{M} \sum_{i=1}^M d_i \,.$$

*Proof.* In the fixed point $\eta_E^\pi(s, a) = \Omega_M \mathcal{T}^\pi \eta_E^\pi(s, a)$, the distributional error estimated by $w_{avg}(s, a)$ does not vanish, unlike the ground truth error $w_1(\eta_E^\pi, \Omega_M \mathcal{T}^\pi \eta_E^\pi)(s, a) = 0$. The shown property upper bounds this mismatch and is a direct consequence of the assumption $w_p(\Pi_i\nu, \nu) \leq d_i$ which postulates an upper bound on the error introduced by the projection $\Pi_i$ in terms of the $p$-Wasserstein distance. The average disagreement is given by

$$w_{avg}(s, a) = \tfrac{1}{M(M-1)} \sum_{i,j=1}^M w_p(\hat{\eta}_i, \hat{\eta}_j)(s, a) \,.$$

The proof is given by applying the triangle inequality and the assumption of the proposition with

$$w_p(\hat{\eta}_i, \hat{\eta}_j) = w_p(\Pi_i \eta_E^\pi, \Pi_j \eta_E^\pi)$$
$$\leq w_p(\Pi_i \eta_E^\pi, \eta_E^\pi) + w_p(\eta_E^\pi, \Pi_j \eta_E^\pi)$$
$$\leq d_i + d_j \,.$$

Plugging in and rearranging yields the desired result.

**Lemma 10** *[Projection error of the categorical projection (Rowland et al., 2018) ] For any distribution $\nu \in \mathscr{P}([z_{min}, z_{max}])$ with support on the interval $[z_{min}, z_{max}]$ and a categorical projection as defined in Eq. (5) with $K$ atoms $z_k \in [z_1, ..., z_K]$ s.t. $z_1 \geq Z_{min}$ and $z_K \leq z_{max}$, the error incurred by the projection $\Pi_C$ is upper bounded in the 1-Wasserstein distance by the identity*

$$w_1(\Pi_C \nu, \nu) \leq \big[ \sup_{1 \leq k \leq K} (z_{k+1} - z_k) \big] \,.$$

*Proof (restated).* The proof uses the duality between the 1-Wasserstein distance and the 1-Cramér distance stating

$$l_1(\nu, \nu') = \int_{\mathbb{R}} |F_\nu(x) - F_{\nu'}(x)| dx = \int_0^1 |F_\nu^{-1}(\tau) - F_{\nu'}^{-1}(\tau)| d\tau = w_1(\nu, \nu'),$$

and can be obtained by a change of variables. The $l_1$ formulation simplifies the analysis of the categorical projection, yielding

$$\begin{aligned}
w_1(\Pi_C \nu, \nu) &= \int_{\mathbb{R}} |F_{\Pi_C \nu}(x) - F_\nu(x)| dx \\
&\leq \sum_{k=1}^{K-1} (z_{k+1} - z_k) |F_{\Pi_C \nu}(z_k) - F_\nu(z_k)| \\
&\leq \sum_{k=1}^{K-1} (z_{k+1} - z_k) |F_\nu(z_{k+1}) - F_\nu(z_k)| \\
&\leq \left[ \sup_{1 \leq k \leq K} (z_{k+1} - z_k) \right] \sum_{k=1}^{K-1} |F_\nu(z_{k+1}) - F_\nu(z_k)| \\
&\leq \left[ \sup_{1 \leq k \leq K} (z_{k+1} - z_k) \right].
\end{aligned}$$

**Lemma 11** *[Projection error of the quantile projection (Dabney et al., 2018b) ] For any distribution $\nu \in \mathscr{P}([z_{\min}, z_{\max}])$ with support on the interval $[z_{\min}, z_{\max}]$ and a quantile projection defined according to Eq. (6) with $K$ equally weighted locations $\theta_k \in [\theta_1, ..., \theta_K]$, the error incurred by the projection $\Pi_Q$ is bounded in the 1-Wasserstein distance by the identity*

$$w_1(\Pi_Q \nu, \nu) \leq \frac{z_{\max} - z_{\min}}{K}.$$

*Proof (restated).* The projection $\Pi_Q$ is given by

$$\Pi_Q \nu = \frac{1}{K} \sum_{k=1}^K \delta_{F_\nu^{-1}(\tau_k)}, \qquad \text{where } \tau_k = \frac{2k-1}{2K}.$$

The desired identity $w_1(\Pi_Q \nu, \nu)$ is accordingly given by the continuous integral

$$w_1(\Pi_Q \nu, \nu) = \int_0^1 |F_{\Pi_Q \nu}^{-1}(\tau) - F_\nu^{-1}(\tau)| d\tau,$$

and can be rewritten in terms of a sum of piecewise expectations

$$w_1(\Pi_Q \nu, \nu) = \sum_{k=1}^K \frac{1}{K} \mathbb{E}_{X \sim \nu} \left[ |X - F_\nu^{-1}(\tfrac{2k-1}{2K})| \big| F_\nu^{-1}(\tfrac{k-1}{K}) < X \leq F_\nu^{-1}(\tfrac{k}{K}) \right].$$

From this, it follows that

$$\begin{aligned}
w_1(\Pi_Q \nu, \nu) &\leq \frac{1}{K} (F_\nu^{-1}(1) - F_\nu^{-1}(0)) \\
&\leq \frac{z_{\max} - z_{\min}}{K}.
\end{aligned}$$

**Corollary 12** *Let $\eta_E^\pi(s, a)$ be the fixed point return distribution for an ensemble of the categorical and quantile projections with the mixture operator $\Omega_M \eta(s, a) = 1/2 \, \Pi_Q \eta(s, a) + 1/2 \, \Pi_C \eta(s, a)$. Furthermore, suppose the return distribution $\eta_E^\pi(s, a)$ has bounded support on the interval $(R_{\max} - R_{\min})/(1 - \gamma)$ where $R_{\max}$ and $R_{\min}$ denote the maximum and minimum immediate reward of the MDP. The average ensemble disagreement $w_{avg}(s, a)$ is then bounded by*

$$w_{avg}(s, a) \leq \frac{4(R_{\max} - R_{\min})}{(1 - \gamma)K}.$$

*Proof.* The result follows straightforwardly from Proposition 9 and Lemmas 10, 11.

Table 1: Hyperparameter search space for bsuite

| Hyperparameter | Values |
|---|---|
| Neural net architecture | $[[64, 64], [128, 128], [512]]$ |
| Learning rate | $[5 \times 10^{-5}, 1 \times 10^{-4}, 5 \times 10^{-4}, 1 \times 10^{-3}]$ |
| Prior function scale | $[0.0, 5.0, 20.0]$ |
| Heads $K$ | $[51, 101]$ |
| Initial bonus $\beta$ | $[0.5, 5.0, 50.0]$ |

Table 2: Hyperparameter search space for VizDoom

| Hyperparameter | Values |
|---|---|
| Learning rate | $[1.25 \times 10^{-5}, 2.5 \times 10^{-5}, 3.75 \times 10^{-5},$ $5 \times 10^{-5}, 6.25 \times 10^{-5}, 7.5 \times 10^{-5}]$ |
| Prior function scale | $[1.0, 3.0, 5.0]$ |
| Initial bonus $\beta$ | $[0.05, 0.1, 0.5, 1.0, 5.0]$ |

## A.5 THE CATEGORICAL PROJECTION

The full definition of the categorical (or also Cramér) projection as stated by Rowland et al. (2018) is given below.

**Definition 13** *[Categorical projection (Rowland et al., 2018) ] For a set of fixed locations $z_1, ..., z_K$ where $z_1 < z_2 < ... < z_K$, let $h_{z_k} : \mathbb{R} \to [0, 1]$ be the hat function centered around $z_k$ for $k = 1, ..., K$ given by*

$$h_{z_k}(x) = \begin{cases} \frac{z_{k+1}-x}{z_{k+1}-z_k} & \text{for } x \in [z_k, z_{k+1}] & \text{and } 1 \leq k < K, \\ \frac{x-z_{k-1}}{z_k-z_{k-1}} & \text{for } x \in [z_{k-1}, z_k] & \text{and } 1 < k \leq K, \\ 1 & \text{for } x \leq z_1 & \text{and } k = 1, \\ 1 & \text{for } x \geq z_K & \text{and } k = K, \\ 0 & \text{otherwise}. \end{cases}$$

*Furthermore, let the categorical representation $\mathscr{F}_C$ be defined as a finite mixture of Dirac deltas $\mathscr{F}_C = \{\sum_{k=1}^{K} \theta_k \delta_{z_k} | \theta_k \geq 0, \sum_{k=1}^{K} \theta_k = 1\}$. The categorical projection operator $\Pi_C : \mathscr{P}(\mathbb{R}) \to \mathscr{F}_C$ of a distribution $\nu \in \mathscr{P}(\mathbb{R})$ is then defined as*

$$\Pi_C \nu = \sum_{k=1}^{K} \mathbb{E}_{\omega \sim \nu}[h_{z_k}(\omega)] \delta_{z_k}.$$

## B EXPERIMENTAL DETAILS

We provide a detailed exposition of our experimental setup, including the hyperparameter search procedure, hyperparameter settings, algorithmic details, and the full bsuite experimental results.

## B.1 HYPERPARAMETER SETTINGS

In our experiments, we aimed to keep most hyperparameters between different implementations equal to maintain comparability between the analyzed methods. Algorithm-specific hyperparameters were optimized over a search space of hyperparameters using Optuna (Akiba et al., 2019). The total search space for bsuite and VizDoom are given in Table 1 and Table 2 respectively, where the *Heads K* parameter only applies to distributional algorithms. C51 requires us to define return ranges, which we defined manually and can be found in the online code repository. All algorithms use the Adam optimizer (Kingma and Ba, 2015).

**Bsuite.** For bsuite, the hyperparameter search was conducted on a subselection of environments of the bsuite, as shown in Table 3. For each environment, we evaluate a set of hyperparameters by

Table 3: Hyperparameter search environments

| Environment ID | Horizon in no. of episodes | Scoring function $f$ |
|---|---|---|
| deep_sea/20 | 500 | $\sum_{(s,a)} \mathbb{1}_{\text{visited}}(s,a)$ |
| deep_sea_stochastic/20 | 1500 | $\sum_{(s,a)} \mathbb{1}_{\text{visited}}(s,a)$ |
| mountain_car/19 | 100 | $\sum_0^t (-1)$ |

means of a scoring function. A particular set of hyperparameters is evaluated every $T/5$ episodes with a maximum training horizon of $T$ episodes. The "continuous" scoring functions make the hyperparameter search more amenable to pruning, for which we use the median pruner of Optuna, reducing the computational burden of the combinatorial search space significantly.

Here, $\sum_{(s,a)} \mathbb{1}_{\text{visited}}(s,a)$ is the count of visited state-action tuples and $\sum_0^t (-1)$ is simply the negative number of total environment interactions. For every hyperparameter configuration $\zeta_i$, the scores $f(\zeta_i)$ are calibrated to facilitate a meaningful comparison between different environments. The calibrated score function we use is given by

$$f_c(\zeta_i) = \exp\left(0.693 \frac{f(\zeta_i) - \mu_\zeta}{\sup_i f(\zeta_i) - \mu_\zeta}\right), \tag{15}$$

where $\mu_\zeta$ is the average score of all hyperparameter configurations $\mu_\zeta = \sum_i^N 1/N f(\zeta_i)$, and $\sup_i f(\zeta_i)$ is the maximal score achieved. The calibration function in Eq. (15) was chosen heuristically to have an intuitive interpretation: it assigns a score of 1 to the best-performing hyperparameter configuration, 0.5 to configurations that achieve exact average performance, and decays exponentially according to score. The final score assigned to a hyperparameter configuration $\zeta_i$ is the sum of all scores of the tested environments. Table 4 shows the full set of hyperparameters used for every algorithm.

**VizDoom.** For the VizDoom domain, the hyperparameter search was conducted on the *MyWayHomeSparse-v0* variation with a training budget of 5 million frames, where final configurations were chosen by achieved return at the end of training. Due to the sparsity of the problem, we did not make use of a pruning algorithm. The specific difference between the different variations of the VizDoom environment *MyWayHome* are shown in Fig. 6, where the *sparsity* of the problem is increased by changing the agents spawning location to a room further from the goal position. The network architecture is based to a large extent on the rainbow network proposed by Schmidt and Schmied (2021) who in turn base their architecture on IMPALA (Espeholt et al., 2018). The specific algorithm configuration for VizDoom is given in Table 5 with a schematic of the network architecture shown in Fig. 8. Table 6 shows our preprocessing pipeline used for the VizDoom environments.

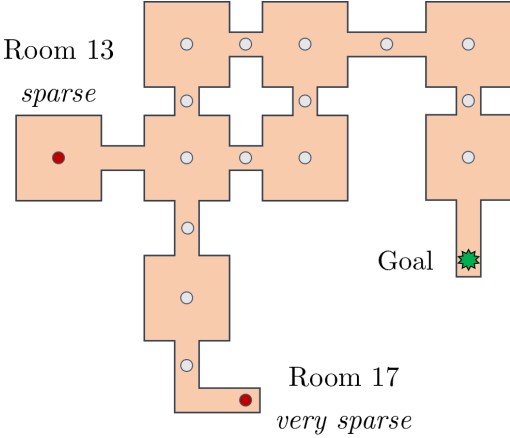

Figure 6: Map for the VizDoom *MyWayHome* environment. Agents are spawned in the *sparse* and *very sparse* locations to vary the exploration difficulty.

## B.2 IMPLEMENTATION DETAILS

**Parametric model.** Our parametric model is a mixture distribution $\eta_{E,\theta}$ parametrized by $\theta$. We construct $\eta_{E,\theta}$ as an equal mixture between a quantile and a categorical representation, each parametrized through a NN with $K$ output logits where we use the notation $\theta_{ik}$ to mean the $k$-th logit of the network parametrized by the parameters $\theta_i$ of the $i$-th model in the ensemble. We consider a sample transition $(s, a, r, s', a')$ where $a'$ is chosen greedily according to $\mathbb{E}_{Z \sim \eta_{E,\theta}(s',a')}[Z]$. Dependencies

Table 4: Hyperparameter settings bsuite

| Hyperparameter | BDQN+P | DLTV | IDS | PE-DQN |
|---|---|---|---|---|
| Net architecture | $[64, 64]$ | $[512]$ | $[64, 64]$ / $[512]$ | $[512]$ |
| Adam Learning rate | $10^{-3}$ | $10^{-3}$ | $10^{-3}$ / $5 \times 10^{-4}$ | $5 \times 10^{-4}$ |
| Prior function scale | 5.0 | 20.0 | 20.0 / 5.0 | 20.0 / 0.0 |
| Heads $K$ | 1 | 101 | 1 / 101 | 101/101 |
| Ensemble size | 20 | 1 | 20/1 | 2/2 |
| Initial bonus $\beta_{\text{init}}$ | n/a | 5.0 | 5.0 | 5.0 |
| Final bonus $\beta_{\text{final}}$ | n/a | n/a | 5.0 | 5.0 |
| Bonus decay (in eps) | n/a | $10^3/N_{\text{episodes}}$ | $0.33 \times N_{\text{episodes}}$ | $0.33 \times N_{\text{episodes}}$ |
| Discount | | | 0.99 | |
| Buffer size | | | $10,000$ | |
| Adam epsilon | | | 0.001/batch size | |
| Initialization | | | He truncated normal (He et al., 2015) | |
| Update frequency | | | 1 | |
| Target update step size | | | 1.0 | |
| Target update frequency | | | 4 | |
| Batch size | | | 128 | |

Table 5: Hyperparameter settings VizDoom

| Hyperparameter | BDQN+P | DLTV | IDS | PE-DQN |
|---|---|---|---|---|
| Adam Learning rate | $2.5 \times 10^{-5}$ | $7.5 \times 10^{-5}$ | $2.5 \times 10^{-5}$ | $6.25 \times 10^{-5}$ |
| Prior function scale | 1.0 | 3.0 | 1.0 | 3.0 |
| Heads $K$ | 1 | 101 | 1 / 101 | 101/101 |
| Ensemble size | 10 | 1 | 10/1 | 2/2 |
| Initial bonus $\beta_{\text{init}}$ | n/a | 0.5 | 0.1 | 5.0 |
| Final bonus $\beta_{\text{final}}$ | n/a | n/a | 0.01 | 0.01 |
| Bonus decay (in frames) | n/a | $10^3/N_{\text{frames}}$ | $0.33 \times N_{\text{frames}}$ | $0.33 \times N_{\text{frames}}$ |
| Loss function | Huber | QR-Huber | Huber/C51 | QR-Huber/C51 |
| Initial $\epsilon$ in $\epsilon$-greedy | | | 1.0 | |
| Final $\epsilon$ in $\epsilon$-greedy | | | 0.01 | |
| $\epsilon$ decay time | | | $500,000$ | |
| Training starts | | | $100,000$ | |
| Discount | | | 0.997 | |
| Buffer size | | | $1,000,000$ | |
| Batch size | | | 512 | |
| Parallel Envs | | | 32 | |
| Adam epsilon | | | 0.005/batch size | |
| Initialization | | | He uniform (He et al., 2015) | |
| Gradient clip norm | | | 10 | |
| Regularization | | | spectral normalization | |
| Double DQN | | | Yes | |
| Update frequency | | | 1 | |
| Target update step size | | | 1.0 | |
| Target update frequency | | | 8000 | |
| PER $\beta_0$ | | | 0.45 | |
| n-step returns | | | 10 | |

Table 6: VizDoom Preprocessing

| Parameter | Value |
|---|---|
| Grayscale | Yes |
| Frame-skipping | No |
| Frame-stacking | 6 |
| Resolution | $42 \times 42$ |
| Max. Episode Length | 2100 |

on $(s, a)$ are dropped for conciseness by writing $\theta_{ik}(s, a) = \theta_{ik}$ and $\theta_{ik}(s', a') = \theta'_{ik}$. The full mixture model $\eta_{E,\theta}$ is then given by

$$\eta_{E,\theta} = \frac{1}{2} \sum_{i=1}^{M=2} \sum_{k=1}^{K} p(\theta_{ik}) \delta_{z(\theta_{ik})}, \quad \text{with} \quad \begin{matrix} p(\theta_{1k}) = \frac{1}{K}, \ z(\theta_{1k}) = \theta_{1k}, \\ p(\theta_{2k}) = \sigma(\theta_{2k}), \ z(\theta_{2k}) = z_k, \end{matrix} \tag{16}$$

where $\sigma(x_i) = e^{x_i} / \sum_j e^{x_j}$ is the softmax transfer function. Consequently, this representation comprises a total of $2K$ atoms, $K$ of which parametrize locations in the quantile model, and the remaining $K$ parametrizing probabilities in the categorical representation. The losses used for each projection method are as provided in the main text.

**Distributional estimation of bonuses.**   For the parametric bonus estimate $b_\phi(s, a)$ we use the same procedure for learning a distributional projection ensemble as with extrinsic rewards. Note that it is not necessary for our method to learn a distributional estimate of the bonus but we find that diverse projection ensembles are good value learners in general and simply reuse the existent function approximation machinery for an intrinsic reward instead of the extrinsic reward. We thus have a model of parameters $\phi$ trained with an alternate tuple $(s, a, w_{\text{avg}}, s', a'_\epsilon)$, where we replaced the immediate reward with the ensemble disagreement $w_{\text{avg}}$ and $a'_\epsilon$ is an exploratory action chosen according to the rule

$$a'_\epsilon = \underset{a \in \mathcal{A}}{\arg\max} \left( \mathbb{E}_{Z \sim \eta_{E,\theta}(s,a)}[Z] + \beta \, b_\phi(s, a) \right), \quad \text{where} \quad b_\phi(s, a) = \mathbb{E}_{B \sim \eta_{E,\phi}(s,a)}[B]. \tag{17}$$

Here, $\beta$ is a hyperparameter to control the policy's drive towards exploratory actions.

**Pseudocode.**   We provide pseudocode for a basic version of PE-DQN where we have simplified details such as the previously described distributional estimation of $b_\phi(s, a)$, prioritized replay, double Q-learning, and prior functions for clarity.

**Randomized prior functions**  are added to all baselines and PE-DQN. Specifically, we add the output of a fixed, randomly initialized neural network of the same architecture as the main net, scaled by a hyperparameter, to the main network's logits. In the case of C51, the prior function is added pre softmax. To the best of our knowledge, DLTV-QR does not use prior functions in its original formulation but we find it to be crucial in improving exploration performance. Fig. 7 (b) shows an experiment assessing the exploration performance of DLTV-QR with randomized prior functions and prior scale 20 (*DLTV [rpf20]*) compared to the vanilla implementation without priors (*DLTV [rpf0]*).

**Information-gain** in our IDS implementation for bsuite is computed in a slightly modified way compared to the vanilla version. Nikolov et al. (2019)  compute the information gain function $I(s, a)$ with

$$I(s, a) = \log\left(1 + \frac{\sigma^2(s, a)}{\rho^2(s, a)}\right) + \epsilon_2,$$

where $\sigma^2(s, a)$ is the empirical variance of BDQN+P predictions, $\epsilon_2 = 1 \times 10^{-5}$ is a zero-division protection, and $\rho^2(s, a)$ is the clipped action-space normalized return variance

$$\rho(s, a)^2 = \max\left(\frac{\text{Var}(Z(s, a))}{\frac{1}{|\mathcal{A}|} \sum_{a \in \mathcal{A}} \text{Var}(Z(s, a))}, 0.25\right). \tag{18}$$

---

**Algorithm 1** PE-DQN

---

1: quantile model with parameters $\theta_1$, target parameters $\tilde{\theta}_1$, and $K$ heads
2: categorical model with parameters $\theta_2$, target parameters $\tilde{\theta}_2$, $K$ heads, and grid $[z_1, \ldots, z_K]$
3: bonus estimation model with parameters $\phi$, and target parameters $\tilde{\phi}$
4: exploration parameter $\beta$, learning rate $\alpha$
5: initialize Buffer $\mathcal{B}$
6: sample initial state $s_0$
7: **for** $t = 0, \ldots, T$ **do**
8:     predict locations $[\theta_{11}, \ldots, \theta_{1K}](s_t, a)$ and probabilities $[\theta_{21}, \ldots, \theta_{2K}](s_t, a)$
9:     $Q(s_t, a) := \frac{1}{2} \sum_{k=1}^{K} \theta_{1k}(s_t, a) \frac{1}{K} + \theta_{2k}(s_t, a) z_k$
10:     predict bonus $b_\phi(s_t, a)$
11:     $a_t \leftarrow \arg\max_{a \in \mathcal{A}} \{Q(s_t, a) + \beta b_\phi(s_t, a)\}$
12:     **for** $j = 0, \ldots, N_{\text{trainsteps}}$ **do**
13:         sample transition tuple $(s_j, a_j, r_j, s'_j) \sim \mathcal{B}$
14:         predict locations $[\theta_{11}, \ldots, \theta_{1K}](s_j, a_j)$ and probabilities $[\theta_{21}, \ldots, \theta_{2K}](s_j, a_j)$
15:         predict target locations $[\tilde{\theta}_{11}, \ldots, \tilde{\theta}_{1K}](s'_j, a)$ and probabilities $[\tilde{\theta}_{21}, \ldots, \tilde{\theta}_{2K}](s'_j, a)$
16:         $Q(s'_j, a) := \frac{1}{2} \sum_{k=1}^{K} \theta_{1k}(s'_j, a) \frac{1}{K} + \theta_{2k}(s'_j, a) z_k$
17:         $a'_j \leftarrow \arg\max_{a \in \mathcal{A}} \{Q(s'_j, a)\}$
18:         mixture target $\tilde{\eta}'_M \leftarrow \frac{1}{2} \sum_{k=1}^{K} \frac{1}{K} \delta_{r_j + \gamma \tilde{\theta}_{1k}(s'_j, a'_j)} + \tilde{\theta}_{2k}(s'_j, a'_j) \delta_{r_j + \gamma z_k}$
19:         quantile loss $l_1 \leftarrow \mathcal{L}_Q(\theta_1, \tilde{\eta}'_M)$
20:         categorical loss $l_2 \leftarrow \mathcal{L}_C(\theta_2, \tilde{\eta}'_M)$
21:         wasserstein distance $r_{\text{intr}} \leftarrow w_1(\sum_{k=1}^{K} \frac{1}{K} \delta_{\theta_{1k}(s_j, a_j)}, \sum_{k=1}^{K} \theta_{2k}(s_j, a_j)) \delta_{z_k}$
22:         bonus estimation target $\tilde{b}' \leftarrow r_{\text{intr}} + \gamma b_{\tilde{\phi}}(s'_j, a'_j)$
23:         bonus estimation loss $l_3 \leftarrow MSE(b_\phi(s_j, a_j), \tilde{b}')$
24:         $[\theta_1, \theta_2, \phi]^T \leftarrow [\theta_1, \theta_2, \phi]^T + \alpha \nabla_{\theta_1, \theta_2, \phi}(l_1 + l_2 + l_3)$
25:     **end for**
26:     execute $a_t$ and store $(s_t, a_t, r_t, s_{t+1})$ in $\mathcal{B}$
27: **end for**

---

Table 7: VizDoom wall clock time comparisons

| Environment | BDQN+P | DLTV | IDS | PE-DQN |
|---|---|---|---|---|
| MyWayHome - Dense | 14h 35m | 14h 22m | 16h 49m | 17h 3m |
| MyWayHome - Sparse | 14h 29m | 13h 49m | 16h 11m | 16h 11m |
| MyWayHome - Very Sparse | 21h 27m | 21h 12m | 23h 3m | 23h 3m |

$\mathrm{Var}(Z(s,a))$ here is the variance of the distributional estimate provided by C51. We replace the clipping in Eq. (18) by adding a small constant $\epsilon_1 = 1 \times 10^{-4}$ to $\mathrm{Var}(Z(s,a))$, s.t.

$$\rho_\epsilon(s,a)^2 = \frac{\mathrm{Var}(Z(s,a)) + \epsilon_1}{\epsilon_1 + \frac{1}{|\mathcal{A}|} \sum_{a \in \mathcal{A}} \mathrm{Var}(Z(s,a))} \, .$$

Fig. 7 (b) shows the effect of clipping as in the vanilla version (*IDS-C51 [clip]*) compared to our variation (*IDS-C51 [noclip]*) on the deep sea environment.

**Intrinsic reward priors** are a computational method we implement with PE-DQN, which leverages the fact that we can compute the one-step uncertainty estimate $w_{\mathrm{avg}}(s,a)$ deterministically from a parametric ensemble given a state-action tuple. This obviates the need to learn it explicitly in the bonus estimation step. We thus add $w_{\mathrm{avg}}(s,a)$ automatically to the forward pass of the bonus estimator $b_\phi(s,a)$ as a sort of "prior" mechanism according to

$$b_\phi(s,a) := b_\phi^{\mathrm{raw}}(s,a) + w_{\mathrm{avg}}(s,a) \, ,$$

where $b_\phi^{\mathrm{raw}}$ is the raw output of the bonus estimator NN of parameters $\phi$. In the VizDoom environment, we follow the default pipeline suggested by Burda et al. (2019b) and subsequent works (Burda et al., 2019a) that normalize intrinsic rewards by a running estimate of its marginal standard deviation.

**Bonus decay** is the decaying of the exploratory bonus during action selection. It is well-known that the factor $\beta$ is a sensitive parameter for UCB-type exploration algorithms, enabling efficient exploration when chosen correctly but simultaneously preventing proper convergence when chosen wrongly. Due to the variety of tasks included in the bsuite and VizDoom, we opted for a fixed schedule by which $\beta$ is linearly decayed to 0.0 over one third of the total training horizon. In the bsuite experiments, we apply this schedule to all tested baselines where applicable and chose the initial $\beta_{\mathrm{init}}$ value according to the hyperparameter search. Since the decay rate is a central part of the DLTV algorithm, we here do not use our linearly deacying schedule but adopt the original decay rate of $\beta = \beta_0 * \sqrt{\log(\alpha t)/\alpha t}$ where $\alpha$ is a scaling parameter.

**Ensembles** and their size are a central parameter in IDS and BDQN+P. For the bsuite experiments, we used a size of 20 as in the implementation by Osband et al. (2020) , who find that increasing the ensemble size beyond 20 did not lead to significant performance improvements on the bsuite. Fig. 7 (a) shows a comparison of the influence of ensemble size in BDQN+P compared to PE-DQN. For VizDoom, we used 10 models in accordance with Nikolov et al. (2019) for their Atari experiments. Here, we follow the original implementations and let the ensembles used in BDQN+P and IDS-C51 (and also PE-DQN) share a network body for feature extraction to save computation.

**Replay buffer** In the VizDoom environment, all our algorithms make use of prioritized experience replay (Schaul et al., 2016).

The **computational resources** we used to conduct the bsuite experiments were supplied by Delft High Performance Computing Centre  (DHPC) and Delft Artificial Intelligence Cluster  (DAIC). We deployed bsuite environments in 16 parallel jobs to be executed on 8 NVIDIA Tesla V100S 32GB GPUs, 16 Intel XEON E5-6248R 24C 3.0GHz CPUs, and 64GB of memory in total. In this setup, the execution of one seed on the entire suite experiment took approximately 38 hours for DLTV, 72 hours for PE-DQN, and 80 hours for IDS. Due to the narrower network architecture of BDQN+P, we in this case parallelized environments over 64 Intel XEON E5-6248R 24C 3.0GHz CPUs, taking approximately 76 hours wall-clock time for the entire suite. In the VizDoom environments, we deployed 32 parallel environments for each agent on the same hardware. In this case, computation for $10 \times 10^6$ took approximately 24 hours per seed per environment and did not differ significantly between any of the tested methods. Table 7 shows the average wall clock time for the VizDoom experiments.

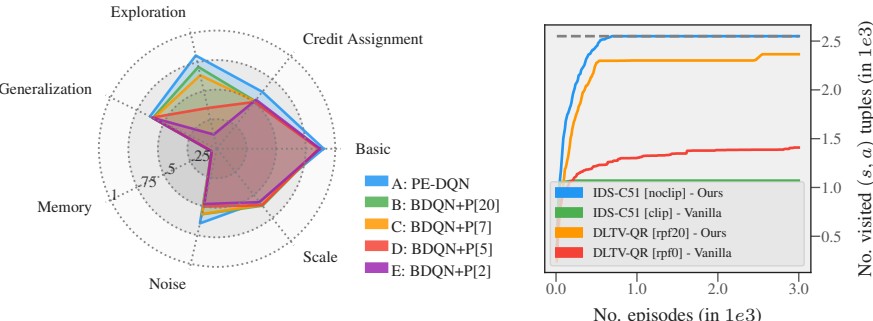

Figure 7: (a) Summary of bsuite experiments. Comparison between BDQN+P with different ensemble sizes and PE-DQN (total ensemble size 4). (b) Deep sea comparison between our implementations and vanilla implementations of baseline algorithms. Shown are median state-action visitation counts over number of episodes on the deep sea environment with size 50. Shaded regions represent the interquartile range of 10 seeds. Higher is better.

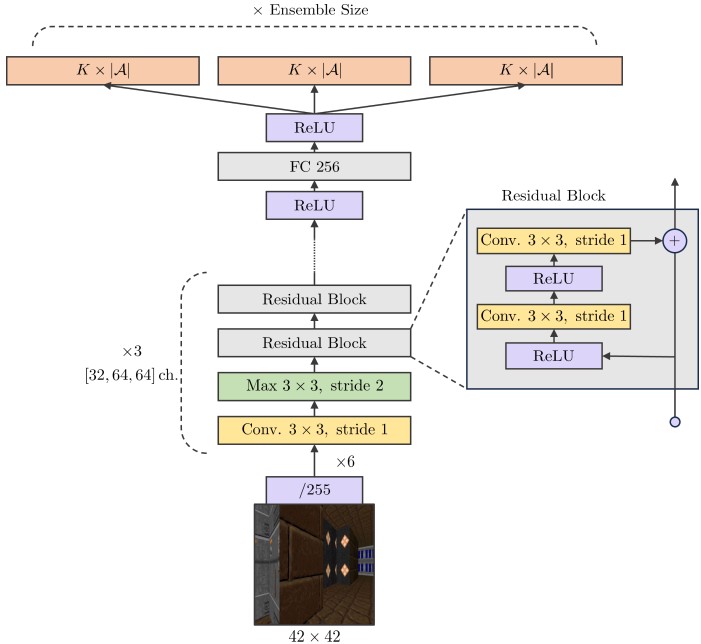

Figure 8: Schematic of the architecture used for VizDoom environments. Based on the architecture used by Espeholt et al. (2018).

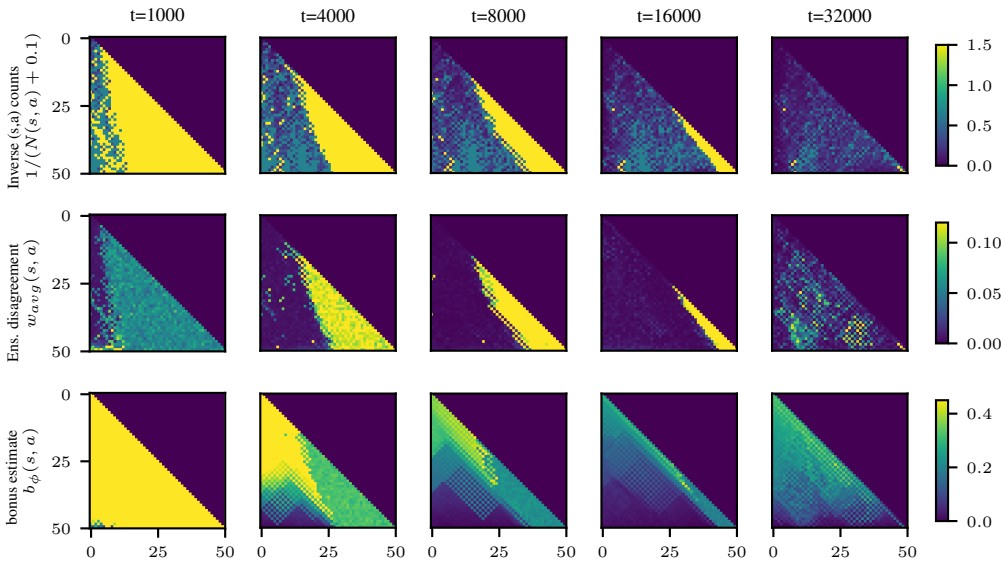

Figure 9: A comparison of inverse counts (top row), ensemble disagreement (mid row), and bonus estimates (bottom row) on the deep sea environment. $t$ indicates total environment interactions. Each image depicts the state-space of deep sea, where only the lower triangle (including the diagonal) is reachable. For each state, the plotted values indicate the maximum of two actions. At $t = 32000$, the agent has discovered the goal-state at the bottom right.

### B.3 ADDITIONAL EXPERIMENTAL RESULTS

Fig. 9 illustrates a comparison of the uncertainty estimates used in PE-DQN for the deep sea environment. Every plot shows the entire state-space of the deep sea environment. In *deep sea*, the agent starts at the top left entry in a matrix and, depending on his action, moves to the left or right column while descending one row. The upper right triangular matrix above the diagonal is thus not reachable to the agent. The goal, i.e., the rewarding final state is located at the bottom right of the matrix.

For different time steps $t$ (total environment interactions) during training, we evaluate the entire state-space and compare three quantities:

- *Inverse counts* are the inverse of visitations to each state-action $\frac{1}{N(s,a)+0.1}$. For every state, we plot the maximum of both actions.

- *Ensemble disagreement* defined as $w_{\text{avg}}(s,a) = 1/(M(M-1)) \sum_{i,j=1}^{M} w_1(\eta_{\theta_i}, \eta_{\theta_j})(s,a)$. For every state, we plot the maximum of both actions.

- *Bonus estimates* $b_\phi(s,a)$ as defined in Section 5.1. For every state, we plot the maximum of both actions.

In the top row, the agent has explored an increasing fraction of the state space with increasing time. The number of states with high inverse counts thus decreases. The ensemble disagreement $w_{\text{avg}}(s,a)$ behaves similarly to inverse counts, a result in line with the notion that $w_{\text{avg}}(s,a)$ serves as an estimate of the local TD error $w_1(\eta_{E,\theta}, \Omega_M \hat{\mathcal{T}}^\pi \eta_{E,\theta})(s,a)$, which is expected to decrease with number of visits. In contrast to this, we expect bonus estimates $b_\phi(s,a)$ to quantify errors w.r.t the true value, that is $w_1(\hat{\eta}, \eta^\pi)(s,a)$. As a result, $b_\phi(s,a)$ should not, for example, vanish prematurely for the initial state at the top left, even after many visitations, since its value can only be assessed upon having explored the entire state space. The bottom row of Fig. 9 is closely in line with this intuition. At $t = 32000$, the agent has discovered the reward at the bottom right.

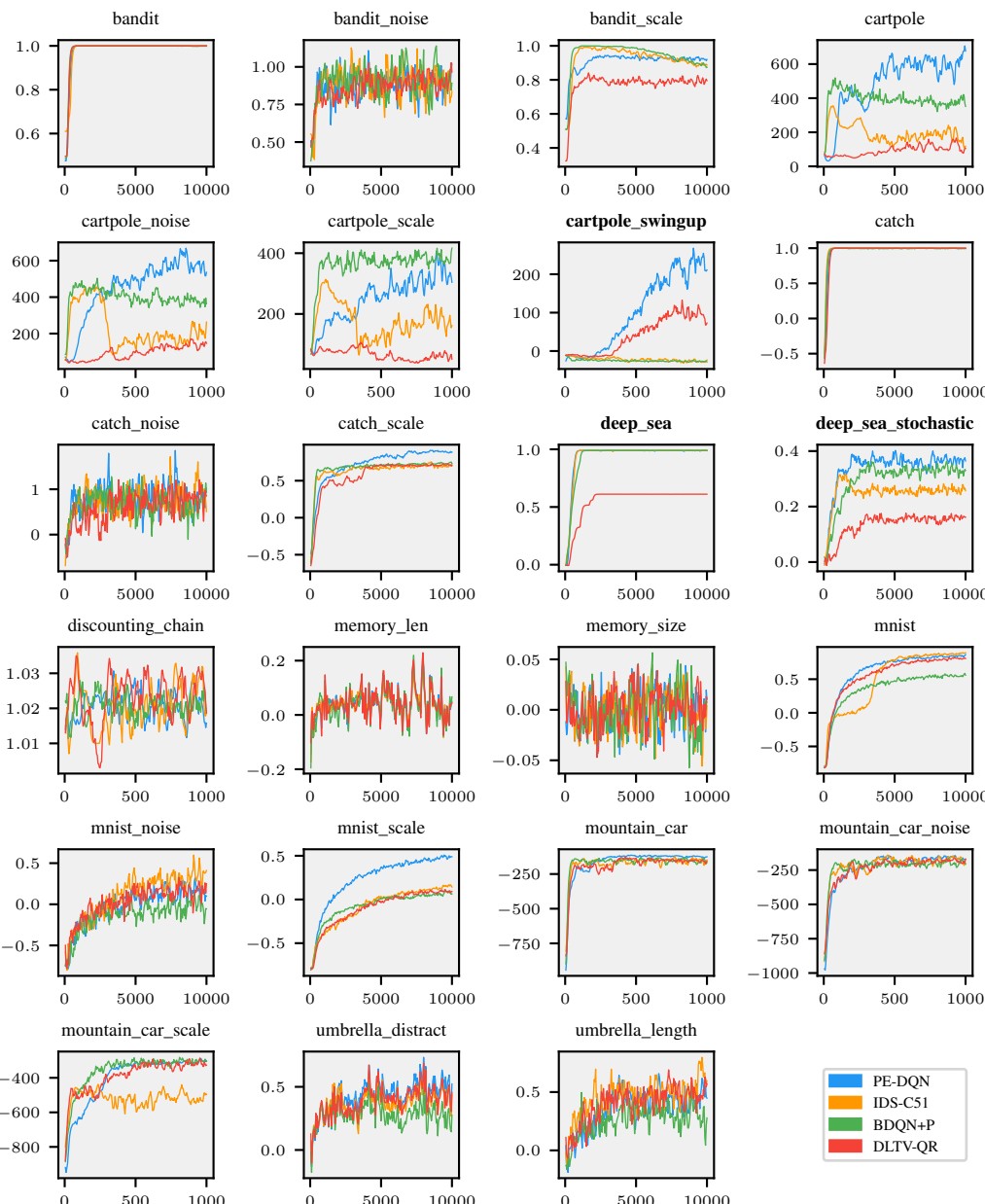

Figure 10: Averaged episodic return for all 23 bsuite tasks.

## B.4 FULL RESULTS OF BSUITE EXPERIMENTS

Fig. 10 shows the averaged undiscounted episodic return for all bsuite tasks. Each curve represents the average over approximately 20 variations of the same task (Osband et al. (2020) provide a detailed account of the task variations) where results were taken from a separate evaluation episode using a greedy action-selection rule. In the "scale" environments, evaluation results were rescaled to the original reward range to maintain a sensible average. Bold titles indicate environments tagged as hard exploration tasks.

