# OpenReview forum: "Diverse Projection Ensembles for Distributional Reinforcement Learning"
_ICLR.cc/2024/Conference — ICLR 2024 poster_

### Official Review · Reviewer_fJQm · 2023-10-19

**Soundness:** 3 good
**Presentation:** 2 fair
**Contribution:** 2 fair
**Rating:** 3
**Confidence:** 5

**Summary:**

This paper proposes projection ensembles method in distributional reinforcement learning for generalization and suggests optimistic bound based on 1-wasserstein distance to promote deep exploration. The authors provide some theoretical results for the proposed Bellman operator. The authors also provide a practical algorithm, PE-DQN and demonstrate its performance on deepmind b-suite benchmark.

**Strengths:**

•	The paper is well-written in general, and easy to follow. The authors provide sufficient background in distributional RL. Their theoretical results were clearly illustrated and the proof appears to be valid. The experimental results of the PE-DQN appear to be promising.

**Weaknesses:**

•	The description of the motivation and related work appears to be rather insufficient.

•	Technical novelty and its contribution seems weak. While the paper focuses on the exploration issue in distributional RL, its theoretical analysis only shows the contractivity and upper bound for proposed Bellman operator.


•	It is a bit odd that the benchmark for comparison does not include the C51, QR-DQN, which employes an $\epsilon$-greedy scheduling method. It is also difficult to verify that the baseline is properly reproduced, since the experiment was not conducted in the standard Atari benchmark.

**Questions:**

•	Minor Comments
o	Section 5.1, paragraph of Uncertainty Propagation : ~exploratory actions.Further details ~ → ~exploratory actions. Further details ~

o	In Section 6.1, $\mathbb{S[Z]}$ should be the mean-variance measure to recover DLTV-QR, not just the variance.

o	In Section 6.2, “the bsuite consists of 22 tasks” seems incorrect as Fig 9 shows 23 bsuite tasks.

o	Appendix B.1, Table 4: DLTV has a fixed decaying schedule, $\sqrt{\logt /t}$. How can $\beta_{final}$ be defined?

o	Appendix B.2, Table 5: Initial bonus $\beta_{init}$ is written twice.

o	Appendix B.2, Equation (17) : What does $\eta_{M,\phi}$ refer to? The expression seems to be ambiguous.

•	- In the paper, an upper bound is obtained by Proposition 2, but it seems insufficient to justify the proposed bonus $b_{\phi}$ as an epistemic uncertainty estimate. Since the confidence bound in Proposition 2 is just an expression naturally induced by the 1-Wasserstein distance, it seems invalid from a regret analysis perspective. Furthermore, the authors again approximate the total distributional error by the one-step TD distributional error, which seems to have no significant theoretical connection. Of course, I think it is quite difficult to describe this theoretically, so I suggest to plot the experimental results of the bonus estimate $\hat{b}_{k+1}(s,a)$ or $\mathbb{E}_{s,a}[\hat{b}_{k+1}(s,a)]$ in each environment.

•	In Figure 4, is PE-DQN [Ind.] without uncertainty propagation equivalent to the C51/QR mixture model with greedy action selection? If so, the performance improvement of PE-DQN seems to be mainly due to the introduction of bonus estimates. Based on the experimental results in the current paper, it is difficult to confirm whether the elimination of the inductive bias by projection contributes to the performance improvement.


•	Projection ensembles seem to be an attempt to relax the priors of the model as in IQN[1] and FQF[2]. Can the authors explain the difference from those?

[1] Dabney, Will, et al. "Implicit quantile networks for distributional reinforcement learning." International conference on machine learning. PMLR, 2018.

[2] Yang, Derek, et al. "Fully parameterized quantile function for distributional reinforcement learning." Advances in neural information processing systems 32 (2019).

---

> ### Author Response · Authors · 2023-11-18
> **Response Reviewer fJQm**
>
> We thank you for the detailed review and useful comments. We hope that you will find our response to some of the raised points and concerns useful.
>
> **Weaknesses**:
>
> - **Motivation and related work**: We thank you for this suggestion and have expanded the related work section: specifically, we added references to Lindenberg et al.[1], Jiang et al.[2], Clement et al.[3], and Parisi et al.[4] with extended discussions. We also made adjustments to our motivating section in the introduction to improve clarity.
>
> - **Technical novelty**: We would like to point out that exploiting an ensemble comprised of models with different projections and representations of potentially disjoint support has several theoretical and practical implications that differ from common ensemble techniques and our results to this end are novel to the best of our knowledge. We are moreover the first to describe a propagation scheme for distributional TD errors to recover upper bounds on total distributional errors and Q-Values (Theorem 3).
>
> - **Additional baselines**: We agree that incorporating experiments with QR-DQN and C51 with $\epsilon$-greedy exploration may provide interesting baselines. However, due to the tight time schedule we instead decided to add additional experiments showing the development of the different uncertainty estimates used in PE-DQN.
>
> **Questions**:
>
> - **Bonus estimation**: We thank you for your question and suggestion to plot the bonuses $b_\phi(s,a)$. To answer your question: Our theoretical results do indeed not merit a full-fledged regret anlysis. We consider this a tall order even for many basic tabular algorithms. However, Proposition 2 proposes a bonus that links total distributional errors to optimistic overestimation of Q-Values, the basis for many furthergoing regret analyses. The connection of this bonus (the total distributional error) to the one-step TD errors is given by the decomposition according to Theorem 3. To illustrate the behavior of this bonus estimate in practice, we have included a Figure in the appendix that plots the bonus estimates $b_\phi(s,a)$ compared to inverse counts and the local error estimates for the entire state-space over several timesteps in the deep sea environment.
>
> - **Ablation experiments**: The referenced section could indeed be clearer and we updated the writing in the current revision. PE-DQN[Ind.] is not equivalent to a C51/QR-DQN mixture model with greedy action selection, but with an optimistic action-selection. I.e., the agent adds a bonus $\beta w_1(\eta_{C51}, \eta_{QR})(s,a)$ to the action-selection, where $\eta_{C51}$ and $\eta_{QR}$ have been trained completely independently, obviating an additional bonus estimation procedure as implied by Theorem 3. Our further ablation experiments, PE-DQN[C51/C51] and PE-DQN[QR/QR], address your second concern of whether the improved exploration performance is in fact due to the diversity of projections in our ensemble, or solely due to the bonus estimation procedure. In this ablation, both PE-DQN[C51/C51] and PE-DQN[QR/QR] make use of the same bonus estimation procedure as PE-DQN, except that ensemble members are based on the same projection and representation.
>
> - **Difference to IQN and FQF**: As you correctly point out, both IQN and FQF, as well as our model, have a higher representational capacity than QR-DQN or C51 in terms of the representable distributions. However, our main focus lies on exploiting the difference in inductive biases and the subsequent difference in generalization behavior, to more reliably quantify epistemic uncertainty in unseen $(s,a)$-regions. Neither IQN nor FQF have this functionality built in and accordingly neither can make use of this in an exploration setting.
>
> - **$\eta_{M,\phi}$**: is a distributional model equivalent in architecture to $\eta_{M,\theta}$ but is trained on intrinsic rewards instead of extrinsic rewards with the goal of estimating the bonus $b_\phi(s,a)$ in the spirit of Theorem 3. Note that we do not require this estimate to be a distributional model per se, but found it convenient to implement given that it equals our procedure for extrinsic return estimation.
>
> **Minor Remarks**:
>
> - We understand that you refer to the "mean-based" variance as opposed to, for example, the "median-based" variance. This would indeed be a more precise term but we believe the definition of variance generally reflects this already.
>
> - We corrected the Section 6.2 to say 23 tasks.
>
> - Indeed, $\beta_{final}$ should be $n/a$ in this entry. We furthermore removed the additional row $\beta_{init}$.

---

> > ### Author Response · Authors · 2023-11-18
> > **References**
> >
> > [1] Lindenberg, Björn, Jonas Nordqvist, and Karl-Olof Lindahl. "Distributional reinforcement learning with ensembles." Algorithms 13.5 (2020).
> >
> > [2] Jiang, Yiding, J. Zico Kolter, and Roberta Raileanu. "On the Importance of Exploration for Generalization in Reinforcement Learning." (2023).
> >
> > [3] Clements, William R., et al. "Estimating risk and uncertainty in deep reinforcement learning."(2019).
> >
> > [4] Parisi, Simone, et al. "Long-Term Visitation Value for Deep Exploration in Sparse-Reward Reinforcement Learning." Algorithms 15.3 (2022).

---

### Official Review · Reviewer_9dG6 · 2023-10-25

**Soundness:** 3 good
**Presentation:** 3 good
**Contribution:** 3 good
**Rating:** 8
**Confidence:** 3

**Summary:**

The paper proposed a novel distributional RL method with improved exploration efficiency. The algorithm maintains an ensemble of return distribution models, each associated with a representation and a projection operator bounded in the p-Wasserstein metric. The ensemble is updated towards a shared TD target computed by first applying the distributional Bellman operator to a return distribution, and then projecting it to individual representations using corresponding projection operator to constitute a new mixture model, an operator the authors named the projection mixture operator. They have demonstrated the contraction property of the proposed projection mixture operator. They then move on to propose to use the W1 distance between true and estimated return distributions as an exploration bonus to construct a provably optimistic Q-value for action selection. Without access to the true return distribution, the authors approximate the exploration bonus with its upper bound which can be estimated incrementally. When the ensemble consists of categorical and quantile projections, the authors provided loss functions on which the model parameters can be updated through gradient descent. Experiments on a variety of tasks have shown improved exploration and overall performance compared to several ensemble / distributional RL benchmarks.

**Strengths:**

1. proposed a novel approach to deep exploration in distributional RL
2. the paper is well written in general with clear and stringent logic
3. has systematically shown that an upper confidence bound of the Q-value can be constructed from the epistemic uncertainty of return distribution estimation in terms of W1, rather than merely treating the latter as a curiosity bonus

**Weaknesses:**

1. since the proposed algorithm is an ensemble on distributional RL, comparisons only with either distributional RL or ensemble on scalar RL but not both seem inadquate. Perhaps worth comparing to some of the works you mentioned in Related Work.
2. the illustration for Fig.1 and Fig.2 can be expanded

**Questions:**

1. is each individual ensemble member constrained to a mixture representation of the return distribution? If so, how is your approach different from a super mixture model containing num_members * num_atoms_per_mixture atoms without splitting into individual mixtures?
2. the atoms in a quantile representation can be thought of with equal weights whereas those in a categorical representation cannot. Would you not need to make this distinction in Eq. 7?
3. the local estimate of the exploration bonus $w_1 (\hat{\eta}, \Omega_M\mathcal{T}^\pi\hat{\eta})$ seems to be measuring the distance between the ensemble return distribution as a whole and its backup target, I failed to see how it may be estimated as the model disagreement among ensemble members (page 7)

---

> ### Author Response · Authors · 2023-11-18
> **Response Reviewer 9dG6**
>
> We thank you for your time, detailed review, and positive recommendation. Please find our reponse to some of the raised points and concerns below.
>
> **Weaknesses**:
>
> - **1. Experimental baselines**: We agree that additional baselines, especially ones that leverage distributional ensembles for exploration would improve the comparative evaluation. The works we reference in Section 2 (e.g., Eriksson et al. [1] and Hoel et al. [2]) are not aimed specifically at exploration and would thus require additional adjustments to make for a sensible comparison in our context. However, other reviewers have pointed towards the recent work by Jiang et al.[3], which should indeed be a sensible comparison. We may consider adding this to the final version, but are afraid the current revision does not allow this due to time constraints.
>
> - **2. Descriptions of Fig. 1 and Fig. 2**: This is a good suggestion and we extended the descriptions for both figures.
>
> **Questions**:
>
> - **1. Mixture representations**: You are correct that the representation of each model in our ensemble is a finite mixture of (Dirac) distributions. Our approach still differes from a "super mixture model" (if we understand you correctly), since each ensemble member individually aims to represent the return distribution as closely as possible (by some metric) within its set of representable distributions. This is not the case for a "super mixture model" with a larger number of parameters. For example, if we construct the uniform mixture $p_M$ of 2 Gaussian mixture models (GMM) $p_1$ and $p_2$ with $N$ parameters, where each GMM fits the same reference distribution, the resulting mixture $p_M$ should be $p_M = p_1 = p_2$. This will not in general be equal to a GMM with $2N$ parameters that fits the same reference distribution. Moreover, unlike an ensemble, a single large mixture model has no built-in way of quantifying epistemic uncertainty.
>
> - **2. Eq. 7**: The distinction between the equal weighting of Diracs in a quantile representation and the varying weights of Diracs in the categorical representation is implicitly contained in $\Pi_i \eta(s,a)$. For the quantile projection, $\Pi_q \eta(s,a)$ yields a mixture of equally weighted Diracs, whereas the categorical projection $\Pi_c \eta(s,a)$ yields a non-uniformly weighted mixture of fixed-location Diracs.
>
> - **3. Local error estimation**: Informally speaking, this procedure is motivated by the fact that each ensemble member in $\eta_{M,\theta}(s,a)$ is trained to regress against the backup target $\Omega_M \hat{\mathcal{T}}^\pi \eta_{M,\theta}(s,a)$. The underlying notion being, that the disagreement of a deep ensemble provides a useful estimate of the expected error between individual predictions and the true labels. This is not a rigorous argument, but rather motivated through the many empirical results of deep ensembles (see Lakshminarayanan et al.[4], Riquelme et al.[5], or Fort et al.[6]), where some works aim to make such statements more formal through connections to the Bayesian setting (e.g., Osband et al.[7], B. He et al.[8], Izmailov et al.[9]). In the current revision, Appendix B.3 contains an illustration of the types of uncertainties used in PE-DQN in practice.
>
> [1] H. Eriksson, D. Basu, M. Alibeigi, and C. Dimitrakakis. Sentinel: Taming uncertainty with ensemble based distributional reinforcement learning. In Uncertainty in artificial intelligence. PMLR, 2022.
>
> [2] C.-J. Hoel, K. Wolff, and L. Laine. Ensemble quantile networks: Uncertainty-aware reinforcement learning with applications in autonomous driving. IEEE Transactions on intelligent transportation systems, 2023.
>
> [3] Jiang, Yiding, J. Zico Kolter, and Roberta Raileanu. "On the Importance of Exploration for Generalization in Reinforcement Learning." arXiv preprint arXiv:2306.05483 (2023).
>
> [4] Lakshminarayanan, Balaji, Alexander Pritzel, and Charles Blundell. "Simple and scalable predictive uncertainty estimation using deep ensembles." Advances in neural information processing systems 30 (2017).
>
> [5] Riquelme, Carlos, George Tucker, and Jasper Snoek. "Deep bayesian bandits showdown: An empirical comparison of bayesian deep networks for thompson sampling." arXiv preprint arXiv:1802.09127 (2018).
>
> [6] Fort, Stanislav, Huiyi Hu, and Balaji Lakshminarayanan. "Deep ensembles: A loss landscape perspective." arXiv preprint arXiv:1912.02757 (2019).
>
> [7] Osband, Ian, John Aslanides, and Albin Cassirer. "Randomized prior functions for deep reinforcement learning." Advances in Neural Information Processing Systems 31 (2018).
>
> [8] He, Bobby, Balaji Lakshminarayanan, and Yee Whye Teh. "Bayesian deep ensembles via the neural tangent kernel." Advances in neural information processing systems 33 (2020): 1010-1022.
>
> [9] Izmailov, Pavel, et al. "What are Bayesian neural network posteriors really like?." International conference on machine learning. PMLR, 2021.

---

> > ### Comment · Reviewer_9dG6 · 2023-11-21
> >
> > Thank you authors for your response. I have no further questions and my positive score persists.

---

### Official Review · Reviewer_a3AN · 2023-10-30

**Soundness:** 3 good
**Presentation:** 3 good
**Contribution:** 3 good
**Rating:** 8
**Confidence:** 3

**Summary:**

This paper considers the use of using ensembles in distributional reinforcement learning. The paper first develops the theoretical framework for using different projection operators in an ensemble and the conditions under which the ensemble still forms a contraction mapping. Then the paper proposes an upper bound on the Q-value estimated by the ensemble via a 1-Wasserstein distance between the learned return distribution, $\hat{\eta}(s, a)$, and the true return distribution $\eta^\pi(s,a)$, $b^\pi(s,a) = w_1(\hat{\eta}, \eta^\pi)(s,a)$. This quantity can be seen as a form of epistemic uncertainty since it would vanish if the learned return distribution is equal to the true return distribution (Proposition 1&2). Since $\eta^\pi(s,a)$ is unknown and uncertainty estimation with bootstrapping can incur bias, the paper then proposes a bellman-like estimate for $b^\pi(s,a)$ that allows it to be estimated recursively using local information (theorem 3).

In the second part of the paper, the paper applies this framework to DQN. The algorithm PE-DQN consists of three components: a QR-DQN ($\theta_1$), a C51 ($\theta_2$), and an uncertainty module ($b_\phi$). During training, $\theta_1$ and $\theta_2$ are updated with the respective distributional RL algorithm, and $b_\phi$ is updated with the uncertainty update from theorem 1. During action selection, the $\theta_1$ and $\theta_2$ are ensembled to estimate the Q-value, and $b_\phi$ is used to estimate the exploration bonus.

Experiments show that PE-DQN is able to outperform relevant baselines in terms of exploration, generalization and credit assignment in bsuite. The paper then shows that in the VizDoom environments with different reward-sparsity levels, PE-DQN is able to match the performance of relevant baselines in the dense reward setting and outperform the baselines in the sparse reward setting.

**Strengths:**

Personally, I really like the ideas proposed in the paper.

-  I find the theoretical motivation compelling and the derivation is straightforward and elegant. The use of 1-Wasserstein between different ensembles provides a nice alternative interpretation of the variance approach many existing works use.
- The proposed uncertainty propagation allows the uncertainty estimation to take in future uncertainty, which many similar methods cannot do, and also addresses some shortcomings of existing methods that do propagate future uncertainties (e.g., the uncertainty bellman equation).
- Empirically, combining the different approaches of distributional RL also makes a lot of sense. Since they have very different inductive biases, it's less likely that they would collapse to the same prediction before the return estimate converges to the true one (in principle).
-  The experiments support the theory reasonably well.

Overall, I think this paper is a nice addition to the distributional RL literature and I am not aware of similar works.

**Weaknesses:**

- The performance improvement over existing methods is relatively small. It would be perhaps ideal to find scenarios where the performance of the PE-DQN is more significant than existing methods. One task that I can think of is Procgen. Recently, [1] showed that incorporating epistemic uncertainty can significantly improve the performance of Value-based methods on Procgen but the method in [1] does not incorporate future uncertainty and only uses an ensemble of QR-DQNs. I would expect PE-DQN to be much better in this case.

- I would like to see a comparison of the wall clock time to get a better understanding of the computational cost.

- I would like to see a sensitive analysis of $\beta$.

- There are several additional works that use ensembles of distributional RL models to do exploration [1, 2, 3]. This paper's contribution is unique (to the best of my knowledge) but I feel it would be better to discuss these works to provide a more comprehensible discussion of what has already been done and what shortcomings this work addresses. [5] also proposes a similar idea but in the tabular setting.

**Questions:**

- I wonder if using different architectures or different ensemble weights for the ensemble would further improve the performance. Perhaps there is even a way where one can adjust the weights to get a tighter estimate of the uncertainty.

- How does this propagation scheme compare to other schemes like UBE [4]?

## Reference

[1] On the Importance of Exploration for Generalization in Reinforcement Learning. Jiang et al.

[2] Distributional Reinforcement Learning for Efficient Exploration. Marvin et al.

[3] Estimating risk and uncertainty in deep reinforcement learning. Clement et al.

[4] The Uncertainty Bellman Equation and Exploration. O'Donoghue. et al.

[5] Long-Term Visitation Value for Deep Exploration in Sparse Reward Reinforcement Learning. Parisi et al.

---

> ### Author Response · Authors · 2023-11-18
> **Response Reviewer a3AN**
>
> We thank you for the insightful review, valuable comments, and positive recommendation. We would like to address some of the points you raised in the following.
>
> **Weaknesses**:
>
> - We agree that it would be very interesting to evaluate PE-DQN on tasks like procgen and that a sensitivity analysis of $\beta$ would be a useful addition. For the current revision this unfortunately seems out of scope due to the computational burden in the limited time.
>
> - We thank you for the useful suggestion and included a wall-clock comparison table for the computationally more demanding VizDoom experiments in the appendix.
>
> - **Related Work** We thank you for pointing out relevant works in this context. We incoporated references to [1],[2] and [3] in the related work section to the best of our ability where trade-offs with the page limit allowed it.
>
> **Questions**:
>
> - **Adjusting weights**: Indeed, this is an interesting avenue for future work. For example, one may define an operator that is "greedy" in its model weighting w.r.t. some criterion (e.g., TD loss of each model) to obtain a mixture that achieves a "best of both worlds" behavior. Similarly, as you suggest, one may consider weighing models to obtain a tighter uncertainty estimate. Intuitively, this should make sense in situations where some models are significantly less reliable than others, where an equal weighting may be overly conservative. The central challenge in this, we suspect, is to find a well-calibrated criterion that does not sacrifice the reliability of uncertainty estimates.
>
> - **Comparison to UBE**: There are a few conceptual differences between our propagation scheme and the uncertainty Bellman equation (UBE) (O'Donoghue et al.), with the clearest being that our bound addresses uncertainty in a distributional sense, whereas the UBE quantifies uncertainty in expected returns. Moreover, our propagation scheme is motivated by a more model-free derivation that directly decomposes total (distributional) errors into distributional one-step TD errors, whereas O'Donoghue et al.[4] decompose the posterior variance of value estimates into posterior variances of reward and transition models. Nonetheless, there are several similarities and perhaps a Bayesian treatment of distributional value estimation may make the links and differences clearer, for example it could be interesting to analyze the tightness of the underlying bounds for these propagation schemes.
>
> [1] Jiang, Yiding, J. Zico Kolter, and Roberta Raileanu. "On the Importance of Exploration for Generalization in Reinforcement Learning." arXiv preprint arXiv:2306.05483 (2023).
>
> [2] Clements, William R., et al. "Estimating risk and uncertainty in deep reinforcement learning." arXiv preprint arXiv:1905.09638 (2019).
>
> [3] Parisi, Simone, et al. "Long-Term Visitation Value for Deep Exploration in Sparse-Reward Reinforcement Learning." Algorithms 15.3 (2022): 81.
>
> [4] O’Donoghue, Brendan, et al. "The uncertainty bellman equation and exploration." International Conference on Machine Learning. 2018.

---

> > ### Comment · Reviewer_a3AN · 2023-11-19
> >
> > I thank the authors for the response and revision of the paper. I have read the other reviews and respective responses. I still think the paper is valuable and will keep my score.

---

### Official Review · Reviewer_1pzE · 2023-11-04

**Soundness:** 2 fair
**Presentation:** 1 poor
**Contribution:** 2 fair
**Rating:** 8
**Confidence:** 4

**Summary:**

This paper proposes a simple ensemble method for distributional RL. The authors discuss some basic theoretical properties of this ensemble and experimentally test its performance.

**Strengths:**

The paper presents a neat and simple idea -- creating a distributional RL algorithm whose return distribution function is an average of multiple return distribution functions. This idea could be easily applied by many practitioners to improve performance. The authors have demonstrated that the ensemble performs better (as with all RL algorithms, the true test of whether it is indeed actually better will be if it is adopted by more people) on bsuite, deep sea environment and VizDoom environment as compared to baselines.

**Weaknesses:**

**Originality**: The paper states that "our work is the first to study the combination of different projection operators and representations in the context of distributional RL", but consider the paper [_Distributional Reinforcement Learning with Ensembles_ by Lindenberg, Nordqvist and Lindahl](https://arxiv.org/abs/2003.10903) from 2020. Specifically, equation (7) in that paper has the exact same form as the formulation of $\eta_M$ in the paper under review.

**Poor writing**: The Achilles heel of this paper is the muddled presentation of ideas. Let me elaborate
1. There are many places where **imprecise writing** makes the presentation difficult to parse. For example, in the abstract it is stated that "profounding impacting the _generalization behavior_ of learned models". What does generalization behavior here mean? It is [known](https://arxiv.org/abs/2111.09794) that generalization in the context of RL can have many different meanings. As another example, in the introduction it is stated that "the space of potential return distributions is infinite-dimensional". The space of probability measures (as opposed to finite signed measures) is _not_ a vector space, and so how do the authors define a dimension in this context? It couldn't be the cardinality of the set because then the set of representable distributions will also have infinite dimensions. Or do the authors mean parametric vs nonparametric, where the dimension is defined on the parameter space? As another example, Figure 1 is poorly explained -- what are the x and y axis, how does it convey the notion of "distinct generalization signature" as stated in the paper? As another example, in section 3, the notation is inconsistent. $S_t$ and $A_t$ are used to denote samples or random variables? If samples, then it is inconsistent with what is stated next that capital letters denote random variables. If random variables, then what does $\mathcal{R}(\cdot \mid S_t, A_t)$ or $\mathcal{R}(S_t, A_t)$ (two ways to stating the same thing?) even mean? $\mathcal{D}^\pi(s,a)$ is in $\mathscr{P}(\mathbb R)$ and not in $\mathscr{P}(\mathbb R)^{\mathcal S \times \mathcal A}$. In the definition of a pushforward measure in the appendix, it is stated that $B \in \mathbb R$, which should be $B \in \mathcal{B}(\mathbb R)$ where $\mathcal{B}(\mathbb R)$ denotes the Borel subsets of $\mathbb R$.
2. There are many places where the presentation is **incomplete or even wrong**. The notion of inverse CDF is not defined anywhere despite it playing a critical role in the paper. When defining the supremum $p$-Wasserstein metric, the authors let $p \in [0, \infty)$. I am pretty sure that this is incorrect and should be $p \in [1, \infty)$. As a simple test, think what happens when $p=0$. See the monograph _One-Dimensional Empirical Measures, Order Statistics, and Kantorovich Transport Distances_ by Bobkov and Ledoux for more details.  The reference of Rösler 1992 for distributional RL makes no sense -- yes it studies related math, but it is not on distributional RL! It is stated in section 3.2 that $\Pi_C$ is a projection defined on $\mathscr{P}(\mathbb R)$, but the complete definition is never even given. It is defined on finite mixtures of Dirac measures, but what about countable mixtures of Dirac measures, or non-atomic measures? In section 4, it is stated that $\eta_M$ has support over $\mathscr{F}_M$. I believe this is incorrect. Consider $\mathscr{F}_1 = \{Z\}$ and $\mathscr{F}_2 = \{-Z\}$, where $Z$ is, say, some random variable having Gaussian distribution. Then $\eta_M = \delta_0 \notin \mathscr{F}_M$.

I really wanted to like this paper -- the idea is simple and potentially useful. But the poor presentation forces me to not recommend acceptance of this paper, unless significantly rewritten. I wasted many hours disentangling, rather than learning, the ideas in the paper, and I wouldn't wish that on potential readers.

**Questions:**

Apart from the multiple questions I asked in the weaknesses section, what is the use of Theorems and Propositions in Section-4, when they play no role in the sections ahead. If they are not empirically useful results, but simply theoretical curiosities, I still don't know what I learned from them. They seem simple manipulations of the various definitions without an end goal in sight.

---

> ### Author Response · Authors · 2023-11-18
> **Response Reviewer 1pZE**
>
> We thank you for your thorough review and value your feedback and comments. We incorporated many of your suggestions and would like to address some of the points you raised and hope you will find our response insightful.
>
> **Weaknesses**:
>
> - **Originality**: We thank you for pointing us to the work by Lindenberg et al.[1] which is indeed relevant to ours and we have accordingly incorporated it in the "Related Work" section of the revised draft. For clarification: in the referenced paper, Eq. (7) is the general definition of a mixture distribution and not a surprising expression to find in the literature. While the work by Lindenberg et al. considers mixtures of distributional models, it does not consider the combination of different projections or representations and unlike our work does not exploit this for efficient exploration.
>
> - **Generalization** We acknowledge that the term "generalization" is not precise and we adjusted our wording in the revision. Intuitively, we considered the term "generalization behavior" to mean a model's predictions on data-points not contained in the dataset. This coincides closely with the description by Kirk et al.[2] for online learning in singleton MDPs (see p. 203 ¶ Scope). If models of an ensemble generalize differently, ensemble disagreement can then be used to quantify epistemic uncertainty in novel inputs.
>
> - **"the space of [...] probability distributions is infinite-dimensional"**: This phrase is indeed does not contribute to clarity and we accordingly replaced it in the revised version. Our notion behind this statement was that one may aim to characterize a probability distribution, for example, through the sequence of its moments. Here, an infinite-dimensional vector containing moments may characterize almost arbitrary distributions. Still, even in this kind of embedding further restrictions would be required, hence our removal of the phrase.
>
> - **Description Fig. 1**: We agree that the description of Fig. 1 benefits from an extension and changed it.
>
> - **Corrections and Notation**: We thank you for pointing out several issues, most of which we have incorporated into the draft: The expression $\mathcal{R}(S_t, A_t)$ should indeed be $\mathcal{R}(\cdot|S_t, A_t)$. $S_t$ and $A_t$ indicate variables as stated and $\mathcal{R}(\cdot|S_t, A_t)$ denotes the reward distribution conditioned on the random variables $S_t$ and $A_t$ as opposed to the explicit events $S_t=s, A_t=a$ where $s$ and $a$ are known to have occured. We furthermore adjusted $\mathcal{D}(Z^\pi(s,a))\in \mathscr{P}(\mathbb{R}^{\mathcal{S}\times\mathcal{A}})$ to $\mathcal{D}(Z^\pi(s,a))\in \mathscr{P}(\mathbb{R})$ and updated the definition of the pushforward measure to read $B \in \mathcal{B}(\mathbb{R})$ where $\mathcal{B}(\mathbb{R})$ are the Borel subsets of $\mathbb{R}$, as suggested. You are correct that $p \in [1,\infty)$ for the $p$-Wasserstein distances and we changed this accordingly. We changed the passage "$\eta_M$ [...] has support over [...]" to express the set of representable mixtures $\mathscr{F}_E = \{ \eta_E(s,a) \,|\, \eta_E(s,a) = \frac{1}{M} \sum_i \eta_i(s,a), \, \eta_i(s,a) \in \mathscr{F}_i, \, i\in[1,...,M]\}$. We respectfully disagree with your counterexample, as $\mathscr{F}_1$ and $\mathscr{F}_2$ are defined to be sets of representable distributions, not random variables. The mixture distribution then is the convex combination of two normal distributions rather than the distribution of the sum of the random variables as implied in your example.
>
> - **Complete definition of $\Pi_C$**: We also agree that including a full definition of the categorical projection $\Pi_C$ is useful and we included this in the appendix.
>
> **Questions**:
>
> - **Clarification of propositions and theorems in Section 4**: We would like to clarify the implication of our results in Section 4. Based on the notion that the choice of projection influences the generalization behavior (in the above-described sense) of a distributional model, we hypothesize, an ensemble that combines different projections can more reliably quantify epistemic uncertainty in unseen $(s,a)$ regions through the disagreement of ensemble members. However, even under the assumption that ensemble disagreement estimates errors between predictions and true targets, disagreement would not yield a useful exploration bonus since we use TD targets not the true return distribution (a known issue addressed for example by Flennerhag et al.[3], O'Donoghue et al.[4], and Fellows et al.[5]). Instead, a bonus that would allow for overestimation of $Q$-values is stated in Proposition 2 and can be decomposed into more accessible distributional TD errors according to Theorem 3. We respectfully disagree that these results are not empirically useful, as they are explicity used in our algorithm and showed consequential in ablation studies in Fig. 4b) rhs. We have also included illustrations in the appendix B.3 that compare these types of uncertainty.

---

> > ### Author Response · Authors · 2023-11-18
> > **References**
> >
> > [1] Lindenberg, Björn, Jonas Nordqvist, and Karl-Olof Lindahl. "Distributional reinforcement learning with ensembles." Algorithms 13.5 (2020): 118.
> >
> > [2] Kirk, Robert, et al. "A survey of zero-shot generalisation in deep reinforcement learning." Journal of Artificial Intelligence Research 76 (2023): 201-264.
> >
> > [3] Flennerhag, Sebastian, et al. "Temporal difference uncertainties as a signal for exploration." arXiv preprint arXiv:2010.02255 (2020).
> >
> > [4] O’Donoghue, Brendan, et al. "The uncertainty bellman equation and exploration." International Conference on Machine Learning. 2018.
> >
> > [5] Fellows, Mattie, Kristian Hartikainen, and Shimon Whiteson. "Bayesian bellman operators." Advances in Neural Information Processing Systems 34 (2021): 13641-13656.

---

> ### Comment · Reviewer_1pzE · 2023-11-21
> **Reply to authors**
>
> Thank you for the reply.
>
> 1. The definition of $\mathscr{F}_E$ now makes sense at least technically.
> 2. The example I had given was stated in terms of random variables, but let me state it in terms of distributions if it helps the authors understand my point. Let the notation $\mathscr{L}(X)$ denote the law (or distribution) of a random variable $X$. That is, if $X \colon \Omega \to \mathbb{R}$ is a random variable defined on a probability space $(\Omega, \mathcal{F}, \mathbb{P})$, let $\mathscr{L}(X) := \mathbb{P} \circ X^{-1}$ be the pushforward measure. Now let $\mathscr{F}_1 = \\{\mathscr{L}(Z)\\}$ and $\mathscr{F}_2 = \\{\mathscr{L}(-Z)\\}$, where $Z$ is a standard Gaussian random variable. Then the only possibility for various $\eta$'s is $\eta_1 = \mathscr{L}(Z)$, $\eta_2 = \mathscr{L}(-Z)$, and $\eta_M = \delta_0$ (here $\delta_0$ is the Dirac measure on $0$). With the new definition of $\mathscr{F}_M$, $\eta_M \in \mathscr{F}_M = \\{\delta_0\\}$, but this wasn't true for the old definition.
> 3. The definition of the projection operator $\Pi_C$ given in Appendix A.5 doesn't make sense. Suppose $\nu$ is a probability measure supported on $(z_K, \infty)$. Then $\mathbb{E}[h_{z_k}(\omega)] = 1$ for each $k$. This means $\Pi_C \nu$ is not a _probability_ measure anymore.
> 4. I still don't understand the importance of the theoretical results. They seem very forced to me.
> 5. I agree that ensemble methods can help with uncertainty. But I don't see big enough improvements with the paper's algorithm.

---

> > ### Author Response · Authors · 2023-11-22
> > **Response Reviewer 1pZE**
> >
> > We thank you for your swift response and would like to address some of the points you raised:
> >
> > 2. In our work, we follow the general definition of finite mixtures, i.e. in the example given here $\eta_E = \frac{1}{2} \eta_1 + \frac{1}{2} \eta_2 = \frac{1}{2} \mathcal{L}(Z) + \frac{1}{2} \mathcal{L}(-Z)$. One may also interpret this as $\eta_E = \mathcal{L}(Y)$ where $Y=Z$ with probability $0.5$ and $Y=-Z$ with probability $0.5$. This mixture can only equal the Dirac measure located at $0$, $\delta_0$, if both $\eta_1 = \delta_0$ and $\eta_2 = \delta_0$. The given example, on the other hand, implies that $\eta_E = \mathcal{L}(Z + (-Z))$, which would indeed result in $\eta_E = \delta_0$.
> >
> > 3. We changed the choice of locations to require $z_1 < z_2 < ... < z_K$ instead of $z_1 \leq z_2 \leq ... \leq z_K$ to prevent degenerate models where $z_1 = z_2 = ... = z_K$. We hope that this answers the point raised by the reviewer.

---

> > > ### Comment · Reviewer_1pzE · 2023-11-22
> > > **Reply to authors**
> > >
> > > 2. I see. It makes sense now. Thank you.
> > >
> > > 3. I still don't see how to fixes the problem. Because of the way $h_{z_k}$ is defined, its expectation will always be $1$ if $\nu$ is supported on $(z_K, \infty)$. This will give $\Pi_c \nu = \sum_{k=1}^K \delta_{z_k}$, which is not a probability measure.

---

> > > > ### Author Response · Authors · 2023-11-22
> > > > **Response Reviewer 1pZE**
> > > >
> > > > Thank you again for the swift response.
> > > >
> > > > In the described case, where $\\nu$ has support over $(z_K, \\infty)$, we have only for $h_{z_K}$ that $E_{\\omega \\sim \\nu} [h_{z_K}(\\omega)] =1$.
> > > >
> > > > For $k=K-1$, the definition of $h_{z_{K-1}}(x)$ reduces to
> > > > \begin{align*}
> > > >     h_{z_{K-1}}(x) &= \\begin{cases}
> > > >              & \\frac{z_{K}-x}{z_{K}-z_{K-1}} \\quad \\text{for } x \\in [z_{K-1}, z_{K}],  \\\\
> > > >              & \\frac{x-z_{K-2}}{z_{K-1}-z_{K-2}} \\quad \\text{for } x \\in [z_{K-2}, z_{K-1}],  \\\\
> > > >              & 0 \\quad \\text{otherwise}.
> > > >          \\end{cases}
> > > > \end{align*}
> > > > Because $\\nu$ has support over $(z_K, \\infty)$, we have $E_{\\omega \\sim \\nu}[h_{z_{K-1}}(\\omega)] = 0$. Similarly is $E_{\\omega \\sim \\nu}[h_{z_{k}}(\\omega)] = 0$ for $k = K-2, K-3, ..., 1$, where $h_{z_{1}}(x)$ reduces to
> > > > $$
> > > >     h_{z_{1}}(x) = \\begin{cases}
> > > >              & \\frac{z_{2}-x}{z_{2}-z_{1}} \\quad \\text{for} \\, x \\in [z_{1}, z_{2}],  \\\\
> > > >              & 1 \\quad \\text{for} \\, x \\leq z_1,  \\\\
> > > >              & 0 \\quad \\text{otherwise}.
> > > >          \end{cases}
> > > > $$
> > > > Accordingly, the projection $\Pi_C \nu$ is then given by $\Pi_C \nu = \\sum_{k=1}^K E_{\\omega \\sim \\nu}[h_{z_{k}}(\\omega)] \\delta_{z_k} = \delta_{z_K}$.

---

### Meta-Review · Area_Chair_SETj · 2023-12-09

**Metareview:**

This paper proposes a novel strategy for exploration in distributional reinforcement learning. They propose to use projections of the uncertainty distribution onto simplified distributions as an ensemble to quantify uncertainty in the distribution. They study theoretical properties of different distribution projection techniques, and evaluate the effectiveness of their approach on a benchmark, demonstrating that it outperforms several baselines.

Most of the reviewers agreed that the approach was a compelling solution to an important problem. They liked the elegant and well-motivated theoretical approach, as well as the fact that the experiments support the theory. One reviewer also liked the fact that the uncertainty quantification can account for future uncertainty. There were some concerns about presentation that the authors have worked to address. Finally, one of the reviewers also raised some concerns about potentially missing baselines.

**Justification For Why Not Higher Score:**

While the proposed approach is interesting and novel, there were some concerns about missing baselines in the experiments that remain to be addressed.

**Justification For Why Not Lower Score:**

The paper combines a well-motivated theoretical idea with supportive experimental results.

---

### Decision · Program_Chairs · 2024-01-16

Accept (poster)